# Plasma proteome profiling reveals dynamic of cholesterol marker after dual blocker therapy

Jiacheng Lyu[1,4], Lin Bai [1,4], Yumiao Li[2,4], Xiaofang Wang[2,4], Zeya Xu[1], Tao Ji[1], Hua Yang[2], Zizheng Song[2], Zhiyu Wang[2], Yanhong Shang[2], Lili Ren[2], Yan Li[3], Aimin Zang[2], Youchao Jia [2] ✉ & Chen Ding [1] ✉

Dual blocker therapy (DBT) has the enhanced antitumor benefits than the monotherapy. Yet, few effective biomarkers are developed to monitor the therapy response. Herein, we investigate the DBT longitudinal plasma proteome profiling including 113 longitudinal samples from 22 patients who received anti-PD1 and anti-CTLA4 DBT therapy. The results show the immune response and cholesterol metabolism are upregulated after the first DBT cycle. Notably, the cholesterol metabolism is activated in the disease non-progressive group (DNP) during the therapy. Correspondingly, the clinical indicator prealbumin (PA), free triiodothyronine (FT3) and triiodothyronine (T3) show significantly positive association with the cholesterol metabolism. Furthermore, by integrating proteome and radiology approach, we observe the high-density lipoprotein partial remodeling are activated in DNP group and identify a candidate biomarker APOC3 that can reflect DBT response. Above, we establish a machine learning model to predict the DBT response and the model performance is validated by an independent cohort with balanced accuracy is 0.96. Thus, the plasma proteome profiling strategy evaluates the alteration of cholesterol metabolism and identifies a panel of biomarkers in DBT.

During the last decade, immune checkpoint inhibitors (ICIs) have emerged in the clinical oncology therapy[1]. ICIs target inhibitory receptors such as the programmed cell death protein 1[2] (PD-1, the drug nivolumab and pembrolizumab) or cytotoxic T-lymphocyte associated protein 4[3] (CTLA4, the drug ipilimumab) on T cells, thereby boosting the antitumor immune response. It has been reported that the monotherapy of PD1 or CTLA4 blockade has prolonged the survival of patients with various advanced tumors[4], including melanoma,

lymphoma, renal cell cancer, head and neck squamous cell cancer, liver cancer, lung cancer, and breast cancer.

While the progression of ICIs is impressive, it has to be acknowledged that just a subset of patients with advanced tumors can respond to ICIs monotherapy, whereas most patients would not[5]. Moreover, the immune-related adverse events (irAEs) such as colitis, diarrhea, dermatological toxicity, endocrinopathy, hepatotoxicity and pneumonitis limit the clinical usage of ICIs immunotherapy[6]. For now, DBT has

[1]Center for Cell and Gene Therapy, Fudan University Clinical Research Center for Cell-based Immunotherapy, State Key Laboratory of Genetic Engineering and Collaborative Innovation Center for Genetics and Development, School of Life Sciences, Human Phenome Institute, Shanghai Pudong Hospital, Fudan University, Shanghai 200433, China. [2]Department of Medical Oncology, Affiliated Hospital of Hebei University; Hebei Key Laboratory of Cancer Radiotherapy and Chemotherapy, 212 Yuhua East Road, Baoding, Hebei 071000, China. [3]Department of Haematology, Hebei General Hospital, No. 348, Heping West Road, Shijiazhuang, Hebei 050051, China. [4]These authors contributed equally: Jiacheng Lyu, Lin Bai, Yumiao Li, Xiaofang Wang. ✉e-mail: youchaojia1@163.com; chend@fudan.edu.cn

emerged as an improved strategy to encounter the above issues, manifesting as the increased response rates and the duration of response[5,7,8]. The studies indicate that the DBT such as the combination treatment with anti-PD1 and anti-CTLA4 ICIs have significantly increased the objective response rates (ORRs) in melanoma (from 43.7% to 57.6%)[9] and non-small cell lung cancer (NSCLC) (35.9%)[10]. Clearly, the DBTs are a next logical step in immunotherapy to improve response rates, increase cure rates and the duration of responses.

The QL1706 is a developed dual immune checkpoint blockade containing a mixture of anti-PD1 IgG4 and anti-CLTA4 IgG1 antibodies[11]. QL1706 showed promising anti-tumor activity in multiple solid tumors. A published phase I/Ib trail demonstrated the ORR and median duration of response were 16.9% (79/468) and 11.7 months, respectively[11].

However, there are still some issues need to be addressed in DBT. For instance, the predictive biomarkers have not yet been developed to identify whether the patient will respond to the DBT[5]. In monotherapy, especially anti-PD1 therapy, it is well-established that some of biomarkers including PD-L1 status, microsatellite instability[12], mismatch repair deficiency[4], and lactate dehydrogenase (LDH)[13] showed the strong predictive power before and during the therapy. Regrettably, the role of these biomarkers in the response to DBT has not been extensively studied. The exploring of DBT response markers will greatly promote the clinical immunotherapeutic trails and the further precise medicine. It is expected that integrating immunohistochemistry, clinical indicators and gene expression signatures will improve the predictive biomarker algorithms. It is easier to translate to the clinical application by embedding the gene expression signatures into predictive model owing to its simultaneously quantified ability for a bunch of genes[7].

Many patients treated with ICIs develop some irAEs such as thyroid problems, manifested by a short period of hyperthyroidism followed by hypothyroidism[14]. Thyroid dysfunction during pembrolizumab treatment of NSCLC is characterized by early-onset, frequently preceded by hyperthyroidism, and may be associated with improved outcomes[15]. The combination CTLA4 and PD1 therapy showed the highest incidence of thyroid problems (56%), followed by the anti-PD1 and anti-CTLA4 monotherapies. In addition, the hyperthyroidism was associated with longer progression free survival (HR = 0.68) and overall survival (HR = 0.57)[16]. However, the underlying mechanisms by which elevated thyroid hormones' level led to favorable outcomes and their potential in response of DBT therapy need further investigation.

Hypercholesterolemia is associated with better outcomes in ICI-treated cancer patients[17]. Meanwhile, it has been reported that hypercholesterolemia induces the activation and proliferation of immune cells, including macrophages, neutrophils, and T cells[18], and the intracellular cholesterol homeostasis is an important regulator of immune cell function[19,20]. Furthermore, the peripheral blood cholesterol level could be a biomarker for the efficacy of the immunotherapy[20,21]. Hence, exploring from the perspective of cholesterol-related biological processes is beneficial to elucidate the intrinsic biological mechanism of immunotherapy and predict the response.

Proteins in the circulatory system directly reflect the individual's physiology. Plasma is the dominant sample used for diagnostic analysis in the clinical practice. Recent mass-spectrometry (MS) based proteomics advanced technology enables the in-depth protein identification and stable quantification for biomedical and clinical research, which make it suitable for the study of disease mechanisms, drug efficacy, and the biomarker exploring[22–24]. Based on these advances, it is possible to integrate the proteome clinical, pathology, and radio imaging of the patients who received the immunotherapy[25–27], which will greatly improve the efficiency of the mechanism illustrating and biomarker screening. This strategy enabled us to explore the connection between thyroid hormone and cholesterol metabolism, and its

positive effects in DBT response. In addition, we identified the biological meaningful apolipoprotein as the predictive biomarker with strong clinical relevance. Additionally, by integrating the gold criterion blood routine indicators and ideal biomarker in plasma, incorporating the machine learning algorithm, we could build up a robust model as a predictable tool for the DBT response, and validated its performance in the independent cohort with high accuracy.

## Results

### Cohort characteristics and research design

We firstly made up with a longitudinal DBT cohort in which 22 patients were enrolled for a range from 1st to 29th therapy cycle-long, QL1706-treated (a DBT with anti-PD1 and anti-CTLA4). Among the 22 patients, there are 6 patients with LC (lung cancer), 4 patients with CCA (cholangiocarcinoma), 3 patients with RCC (renal cell carcinoma), 3 patients with OVCA (ovarian cancer), 2 patients with CRCA (colorectal cancer), 2 patients with CESC (cervical squamous cell carcinoma), a patient with BLCA (bladder cancer), and a patient with UCEC (uterine corpus endometrial carcinoma). Totally, 113 samples were collected in the DBT cohort. An independent validation cohort were also collected which consist of 54 longitudinal plasma samples from 27 patients who received anti-PD1 monotherapy. Additionally, we enrolled 24 healthy controls to build the baseline of plasma proteome. In the end, a total of 191 samples were enrolled in this study (Fig. 1A).

For the DBT cohort, all patients were treated according to the standard treatment schedule with medical treatment. Specifically, according to the recommended phase 2 dose of QL1706 dose exploration, 5.0 mg/kg was selected as the treatment dosage. Patients received intravenous infusion every 3 weeks. Radiology imaging was used to evaluate the therapy response by assessing the tumor sites after the QL1706 treatment. According to the iRECIST v1.1 standard, the clinical DBT response was classified as immune complete response (iCR), immune partial response (iPR), immune stable disease (iSD), immune unconfirmed progressive disease (iUPD), and immune confirmed progressive disease (iCPD). Blood samples were collected before each therapy cycle for hematological evaluation, including blood routine, blood biochemistry, coagulation function, myocardial enzyme spectrum, thyroid function, pituitary-adrenal axis, virology (Supplementary Data 1, *Methods*). In the end, 22 and 91 plasma samples were collected from 22 enrolled tumor patients before and during QL1706 DBT, respectively. Among the 91 samples, 2 samples were complete response (iCR), 17 samples were partial response (iPR), 17 samples were stable disease (iSD), 4 samples were unconfirmed progressive disease (iUPD), 7 samples were confirmed progressive disease (iCPD), and 44 samples were not-evaluated (Fig. 1B). We also evaluated the demographic and clinicopathological indicators for patients, including age, gender, BMI, tumor node metastasis (TNM) stage etc. (Table 1, *Methods*). The anti-PD1 cohort was used for validating the ability of the clinical-proteomics biomarker that can distinguish the response to the immunotherapy.

We recently described a highly sensitive and in-depth proteomics sample detection pipeline that can perform an unbiased and rapid "body fluid proteome profiling" following a data-independent acquisition (DIA) based MS strategy[22]. Based on this, we have developed a highly efficient workflow for plasma, which allows the robust measurement in less than 1 h. We speculated this advanced technique should be useful in the analysis of large cohort studies, including the longitudinal monitoring of immunotherapy for patients with advanced tumor. The MaxLFQ algorithm was used for the proteome quantification[28] (*Methods*).

To investigate the quantitative reproducibility of our pipeline, we randomly selected 10 samples from the DBT cohort and detected in consecutive replicates 5 times for each sample (*Methods*). We calculated the coefficient of variance (CV) for each of the sample among the 5 repeats. Fig. S1A showed the CVs of proteins in each of samples (blue

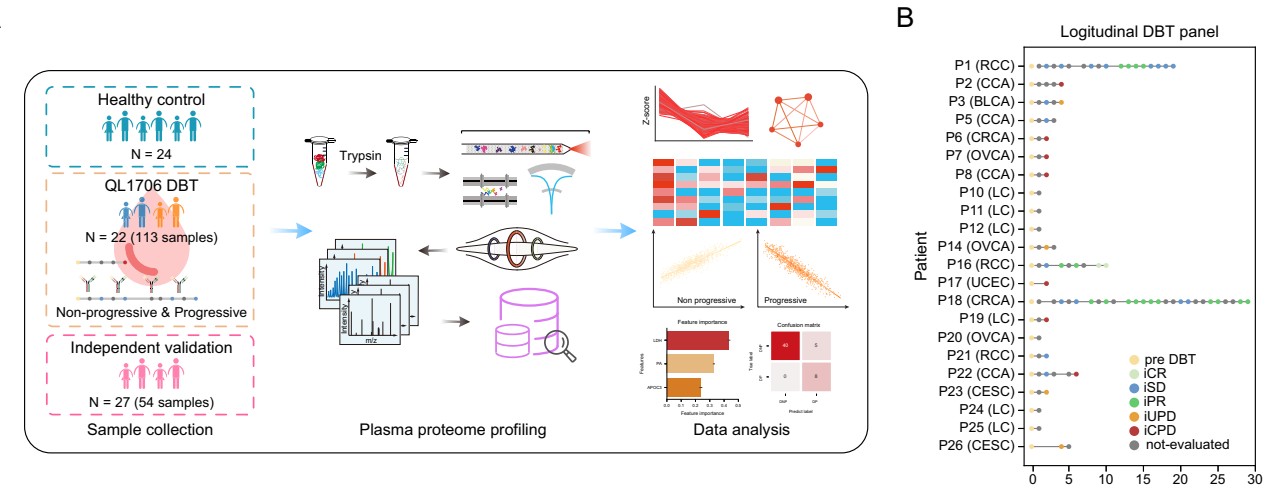

**Fig. 1 | Cohort design and plasma protein profiling pipeline. A** Workflow of bispecific antibody therapy (DBT) cohort and healthy control cohort collection, processing, sequencing, and data analysis. **B** The panel depicting the longitudinal DBT for each patient. The dot represented the evaluated state. Yellow as the pre DBT, light green as complete response, green as partial response (iPR), blue as stable disease (iSD), pink as unconfirmed progressive disease (iUPD), red as confirmed progressive disease (iCPD), and gray as the not-evaluated. Lung cancer (LC), cholangiocarcinoma (CCA), renal cell carcinoma (RCC), ovarian cancer (OVCA), colorectal cancer (CRCA), cervical squamous cell carcinoma (CESC), bladder cancer (BLCA), and uterine corpus endometrial carcinoma (UCEC). Source data are provided as a Source Data file.

dot indicated the CVs less than 30%). In addition, we calculated the Pearson correlation among the 5 repeats for each of samples. As shown in Fig. S1B, the correlation ranged from 0.94 to 0.98. These results suggested the robustness of the MS detection. Therefore, we detected once for each sample in the actual DBT cohort. By applying the advanced liquid chromatography DIA-MS pipeline, we generated a large plasma proteome dataset (Fig. 1A, Supplementary Data 1). To assess the quality of this dataset, we further analyzed the reproducibility of the 15 mixed quality samples (*Methods*). Quantitative accuracy was high as reflected by the high Pearson correlation coefficients with median 0.97 and the median CVs of proteins was 19%, 74% proteins had CV below 30% (Fig. S1C, D), which demonstrated the consistent stability of the MS platform during the DBT samples detection.

As for the proteome dataset, the proteins identified less than 70% of samples were excluded for the downstream analysis (*Methods*). To evaluate the distribution of the dataset, we applied dip test at sample level. As shown in Fig. S2A, the distribution of all samples in DBT cohort were unimodal distribution (dip test $p < 0.05$). Furthermore, we evaluated the batch effects on the proteome dataset by performing the principal component regression analysis (*Methods*). As shown in Fig. S2B, only component 39, component 56, component 77 and component 79 showed significant correlation with batch effects ($p < 0.05$), but with lower explained ratio (less than 1%). These results indicated there were no observable batch effects in this dataset. Further Gene Ontology Biological Process (GOBP) terms annotating revealed statistically significant terms (Fig. S2C). The high abundance proteins were participated in cholesterol metabolism and inflammation processes, whereas the response to stress and phagocytosis were enriched by the low abundance proteins.

## Plasma proteome revealed negative linkage between cholesterol metabolism and oncogenic signaling after DBT

Our initial aim was to investigate the effects of DBT on plasma proteome of patients. Therefore, we compared plasma proteome characteristics among healthy control group, pre DBT group, and 1st DBT cycle group (Fig. 1B). As shown in Fig. 2A, the results revealed dramatic alterations in the proteome composition, manifested as 82 significantly differently expressed proteins (DEPs) (Kruskal−Wallis test FDR < 0.05). We defined the protein expression state as high (H), medium (M), and low (L), and then grouped DEPs into four protein clusters (annotated as HLH, LMH, LHL, and HML) by comparing the relative z-score value among three sample groups (healthy control, pre DBT, and 1st DBT cycle group). For example, in the LMH protein cluster, the healthy control group had low protein levels (L), the pre DBT group had moderate protein levels (M), and the 1st DBT cycle group had high protein levels (H).

Over-represented analysis was performed to explore the enriched biological processes for the four protein clusters (Supplementary Data 2). Proteins in HLH cluster, implying the recovery of tumor-inhibited protein expression after the DBT, participated in lipid binding and antioxidant activity (APOM, CETP, and ALB). LHL protein cluster was characterized by extracellular space (HRG, HGFAC), indicating the "calm down" of tumor-associated proteins after the DBT. LMH cluster showed significantly enrichment of immune-related

## Table. 1 | Baseline demographic and disease characteristics of patients in dual blocker therapy cohort

| Characteristic | Dual Blocker Therapy cohort (*N* = 22) |
|---|---|
| Age -yr. | |
| Median | 55 |
| Range | 35–71 |
| Gender -no. (%) | |
| Female | 7 (32%) |
| Male | 15 (68%) |
| TNM stage -no. (%) | |
| I | 2 (9.1%) |
| II | 1 (4.5%) |
| III | 13 (59.1%) |
| IV | 1 (4.5%) |
| NA | 5 (22.7%) |
| BMI -no. (%) | |
| <18.5 | 0 |
| 18.5–24.9 | 17 (77.3%) |
| >25 | 5 (22.7%) |

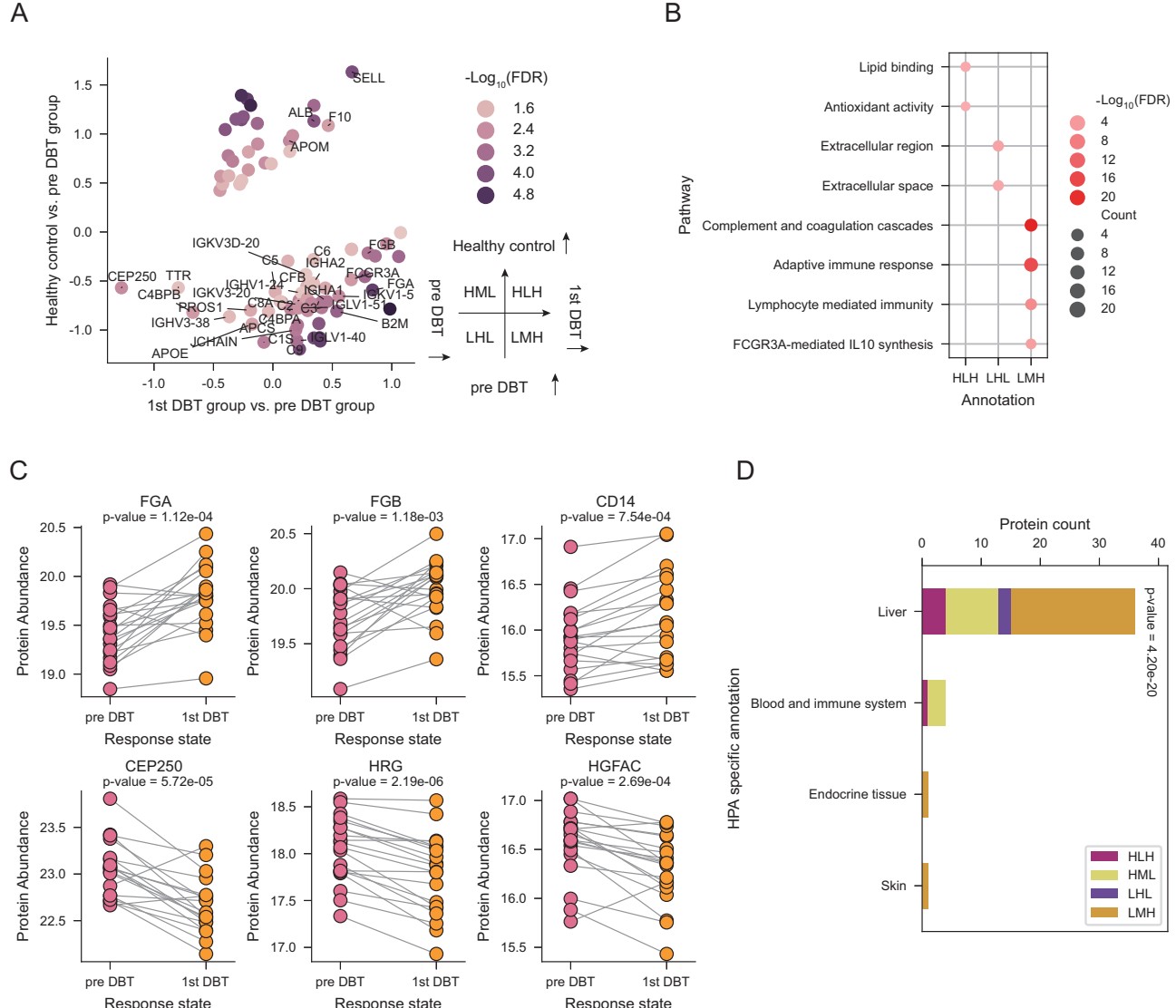

**Fig. 2 | The effects of DBT on plasma proteome. A** Comparison of plasma protein changes before and after DBT treatment. The x-axis indicates the comparison of pre DBT group and first DBT group, and y-axis indicates the comparison of healthy control group and pre DBT group. The color represents the significance of protein (Kruskal-Wallis test false discovery rate (FDR)). **B** The bubble plot depicting the pathway enrichment of three protein groups which defined in Fig. 2A. Dot size represents the protein counts of the pathway and color represents the pathway enrichment significance. **C** Sample-wised plot of proteins involved in the pathway which described in Fig. 2B between pre DBT samples ($n = 20$) and first DBT samples ($n = 20$). $P$ value was derived by two-sided paired $t$ test. **D** The stacked bar plot depicting the overlap of differently expressed proteins and the tissue-specific proteins from The Human Protein Atlas (HPA) dataset. $P$ value was calculated by two-sided chi square test. Source data are provided as a Source Data file.

processes, such as lymphocyte mediated immunity and complement cascade (SAA1, FGA, and CD14) (Fig. 2B, C). These results represented the strong impacts of DBT on the cholesterol processes, immune response, and oncogenic signaling.

ICIs can lead to a distinct constellation of organ-specific side effects[29]. In this study, we wondered whether the DEPs showed the enrichment of organ specific protein expression pattern, which indicated whether the DBT primarily affected the specific organ. Hence, we compared the above four protein clusters with the HPA dataset[30]. As a result, there were 36 DEPs annotated as liver-specific protein (Chi square test $p < 0.05$, Figs. 2D, S3A, Supplementary Data 2), followed by 4 DEPs were annotated as blood and immune system-specific. Meanwhile, we evaluated the level of liver-function indicators between pre DBT group and 1st DBT cycle group samples. The levels of aspartate transaminase (AST), Alanine aminotransferase (ALT), alkaline phosphatase (ALP), and Gamma-glutamyl transferase (GGT), which

reflected the liver function damage[31], were elevated in pairwise samples with 11%, 24%, 13%, and 50% increasement (Fig. S3B). These results indicated that the liver function damage should be considered in DBT clinical trials, and the linkage between DBT and liver function should be further investigated.

### Thyroid hormone related clinical indicators are significantly altered by DBT
In order to deeply illustrate the clinical indicators and proteome characteristics alteration of DBT, we classified iUPD and iCPD samples as disease progressive (DP) group; iPR and iSD as disease non-progressive (DNP) group (*Methods*). The clinical indicators could provide a trace of the biological processes alteration for the physiological system, which aid in understanding the interior changes of DBT. Therefore, we compared the clinical indicators between DP group and DNP group (permutation-based $t$ test FDR < 0.05). As a result, LDH,

alkaline phosphatase (ALP), and white blood count (WBC) were significantly upregulated in DP group and hemoglobin (HGB), red blood cell (RBC), FT3, albumin globulin ratio (A/G), prealbumin (PA, also known as thyroxine-binding prealbumin) level, and Creatine Kinase-MB (CK-MB) were upregulated in DNP group (Fig. 3A). Moreover, we found it only for PA and LDH that the majority of DNP sample levels were within the reference interval, while a part of DP sample levels was not (Figs. 3B, S4A, Supplementary Data 3). This result suggested the abnormal level of PA could indicate the DBT response. In order to confirm the above inference, we tested the significant different clinical indicators including LDH and PA at pair-wised level by comparing the pre DBT group and the DP group. As a result, only PA was significant different in DP samples than matched pre DBT group samples (pairwise $t$ test $p = 0.027$, Figs. 3C, S4B).

PA, also known as thyroxine-binding prealbumin or transthyretin, is a transport protein in the plasma and cerebrospinal fluid that transports the thyroid hormone thyroxine and retinol to the liver[32]. In our cohort, the thyroid hormone FT3 and T3 showed higher level in DNP than DP samples (Fig. 3D, permutation-based $t$ test $p = 1.00E-4$ and $p = 5.40E-3$, respectively). The function of FT3 is similar to thyroxine, but with faster and stronger physiological effects[33]. Furthermore, we found the significantly positive correlation between FT3 and PA (Fig. 3E, pearson corr = 0.50, $p = 1.24E-3$), which indicated attenuation of thyroid hormone synthesis in DP samples. It is well-established that thyroid hormone participates in lipid and glucose metabolism, and the neuron development[33]. To figure out the downstream biological processes of thyroid hormone in DBT response, we calculated the lipid-related, glucose-related, and neuron-related scores by ssGSEA (Methods) and performed pearson correlation analysis with FT3 (Supplementary Data 3). By comparing the correlation coefficients between lipid, glucose, and neuron related processes, we found lipid-related score was most correlated with FT3 (lipid-related vs. glucose-related and neuron-related permutation-based $t$ test $p = 2.00E-4$ and 0.032) (Fig. 3F). In addition, we classified the lipid-related processes as phospholipid, glycerophospholipid, sphingolipid, cholesterol, fatty acid, and ketone bodies. The result showed the cholesterol has the highest correlation with the FT3 (Fig. S4C). This result implied the important role of cholesterol metabolism in DBT response.

### Integrated analysis of proteome and radiology imaging reveals the potential of APOC3 as a biomarker in the DBT response

Having determined the altered clinical indicators between DNP and DP group, we next investigated the differences of plasma proteome profile between two groups. By comparing the ssGSEA scores between two groups (permutation-based t-test FDR < 0.05, Supplementary Data 4), we noticed the lipid-related processes followed by the neuron-related processes were up-regulated in the DNP group (Fig. 4A). The upregulated lipid-related biological processes in DNP group included triglyceride rich lipoprotein particle remodeling, reverse cholesterol transport, lipoprotein lipase activity, lipoprotein particle clearance, and plasma lipoprotein assembly (Fig. 4B). Considering the effect of BMI on the lipid-metabolism[34], we tested the BMI between the two groups and observed there's no significant difference (permutation-based $t$ test $p = 0.60$) (Fig. S5A). Apolipoprotein family members, which can bind and transport blood cholesterol to various tissues of the body for metabolism and utilization, are of central importance in determining the response of immunotherapy[35]. In our cohort, the plasma proteome provided us a comprehensive profile for different apolipoproteins. In detail, the level of high-density lipoprotein constituents APOC3, APOC2, and APOL1 increased in DNP group than DP group (permutation-based $t$ test $p = 2.00E-4$, $1.00E-3$, and $7.70E-3$) (Fig. 4C and Supplementary Data 4). These results implied the alteration of apolipoprotein in DBT.

The above analysis revealed the effects of DBT on the levels of APOC3, APOC2, and APOL1, which implying the potential roles of these proteins for monitoring the DBT response. In order to screen out the valuable proteins, we hypothesized that the ideal biomarker level should be consistently increased or decreased during the DBT for each patient. Therefore, we performed the time-series linear regression for 6 patients (P1, P16, P18 with no DPs and P2, P3, P22 with finally DPs) with more than 3 DBT cycles and set the DBT cycles as the covariable (Methods). Based on the hypothesis, we proposed a criterion to explore biomarkers, as for the patient with no DPs samples, (1) the expression level of the biomarker was gradually increased along with the therapy cycle and (2) the $p < 0.05$ in the linear model. As a result, APOC3 and APOC2 met the criterion (Figs. 4D, S5B). Furthermore, between DNP and DP group, the expression level of APOC3 showed more significant difference than APOC2. Therefore, the APOC3 was selected as the potential biomarker for indicating the DBT response state.

Based on the time-series linear regression, we observed APOC3 level were increased in patients without DP samples and not in patients with DP samples (Fig. 4D). To further validate the potential of APOC3 as the biomarker for DBT response, we assessed the medical radiology imaging for the above patients to comprehensively analyze the relationship between therapy efficacy and APOC3 protein level. For patients without DP samples (P1, P18, and P16), as shown in Fig. 4E, the tumor size of two lung metastases in P16 were gradually decreased, and all tumors disappeared after the 9th treatment (from 15.34 mm and 14.34 mm to disappeared). More importantly, the tumor size of two metastases showed the negative correlation with the APOC3 protein level during the DBT (Fig. S5C). Similar results were shown in P1 and P18 (Fig. S6). In detail, the tumor size of retroperitoneal lymph nodes (from 17.77 mm to 3.76 mm) and liver metastasis (from 21.45 mm and 17.69 mm to 16.74 mm and 10.96 mm) in P18 and kidney metastasis (from 10.81 mm and 7.5 mm to disappeared) in P1 were both negatively correlated with the respective APOC3 protein levels. These results suggested the biomarker potential of APOC3 in patients without DP samples. Oppositely, the patients with decreased APOC3 protein level did not response the DBT (P2 and P3). By evaluating the radiology imaging of these two patients, we observed not only negative association between tumor size and APOC3 protein levels in the patients, but also the appearance some new metastases (Fig. S7). In detail, for P2, APOC3 protein level showed the negative correlation between tumor size of retroperitoneal lymph nodes, porta hepatis, and right pleural metastases. Besides, there are four new metastases appeared, including abdominal lymph nodes, brain, right axillary lymph nodes, and mediastinal lymph nodes. P3 showed a similar pattern, indicating no response to DBT treatment. After analyzing the radiology imaging data, the results further demonstrate the potential of APOC3 as a biomarker for DBT response.

### Integrated proteome-clinical features-based machine learning model provide an accurate measure for DBT

We have identified clinical (PA, LDH) and proteome (APOC3) features present in the longitudinal cohort that associated with the response of DBT. This motivated the use of a machine learning framework (Fig. 5A) to integrate features into a predictive model of DBT response to predict the disease progression state of patient. There were five optionally different feature combinations based on the clinical and proteome features, including (1) PA, (2) LDH, (3) APOC3, (4) integrated clinical features (PA and LDH), (5) integrated all features (PA, LDH, and APOC3). Before the model construction, the collinearity of features was evaluated by the spearman analysis (Fig. S8A).

The models were based on multi-step processing pipeline (Fig. 5A). Inside the pipeline, for each feature combinations, we first compared different data preprocessing strategy (including the StandardScaler, MinMaxScaler, RobustScaler, and Normalizer) along with 21 state-of-the-art machine learning models and selected the top N (such as $N = 5$) best models. Next, a threefold cross-validation scheme was used to optimize model hyperparameters and the best model was

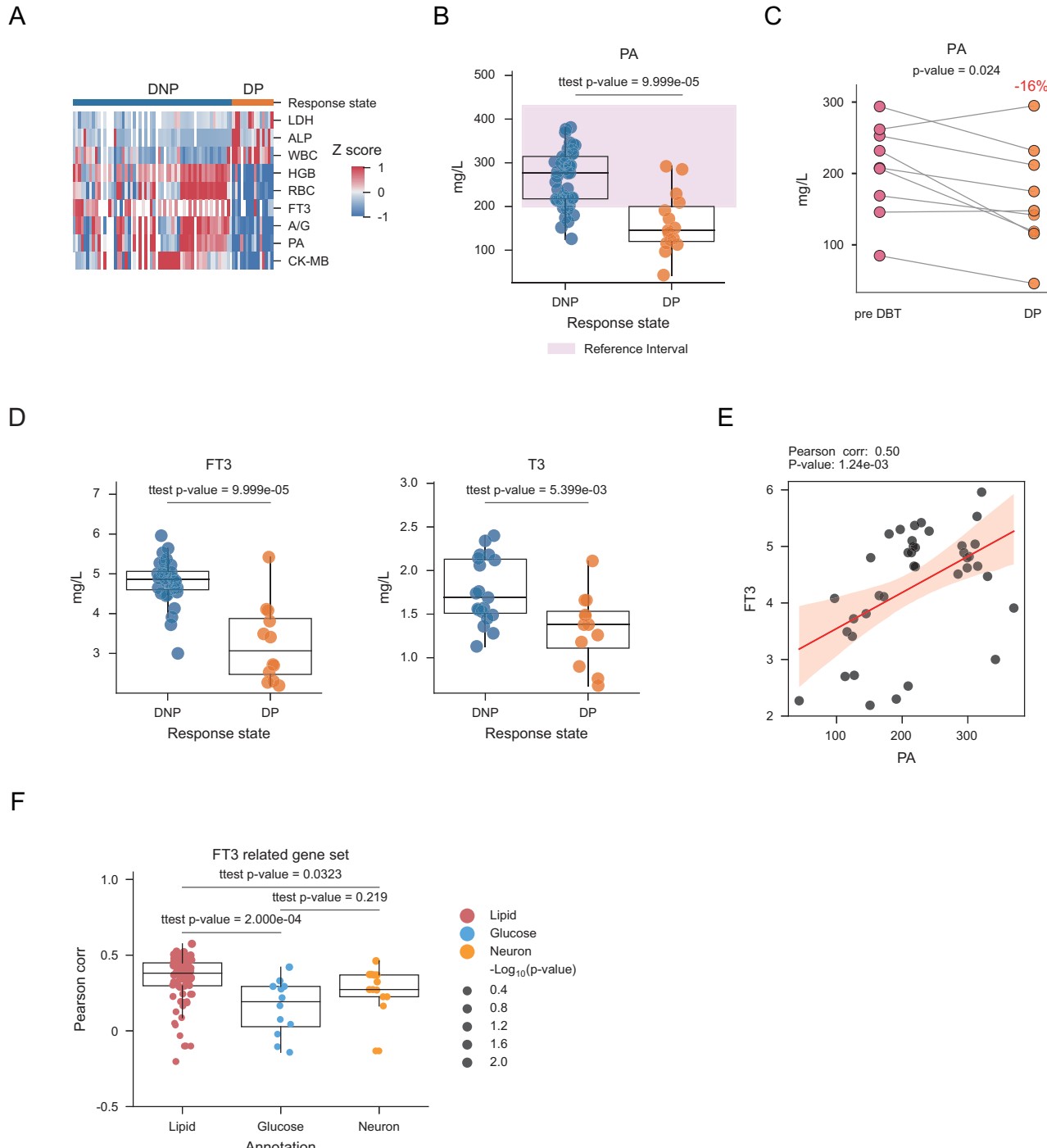

**Fig. 3 | Thyroid hormone related processes were activated in DBT. A** Heatmap depicting the significant different level of blood routines. FDR were derived from the adjusted two-sided permutation-based *t* test *p* values. Values were transformed by z-score. **B** The boxplot describes the level of prealbumin (PA) between disease non-progressive (DNP) (*n* = 47) and disease progressive (DP) (*n* = 15) samples at blood routine level. The pink rectangle shows the normal reference interval. *P* value was derived by two-sided permutation-based *t* test. The box ranges from the first (Q1) to the third quartile (Q3) of the distribution and represents the interquartile range (IQR). A line across the box indicates the median. The whiskers are lines extending from Q1 and Q3 to end points that defined as the most extreme data points within Q1 − 1.5 × IQR and Q3 + 1.5 × IQR, respectively. **C** Pair-wised plot of PA at blood routine level between pre DBT samples (*n* = 9) and DP samples (*n* = 9). *P* value was derived by two-sided paired *t* test. **D** The boxplot describes the free triiodothyronine (FT3) and the triiodothyronine (T3) level of between DNP (*n* = 32 for FT3 and *n* = 19 for T3) and DP samples (*n* = 12 for both FT3 and T3) at blood routine level. The box ranges from the Q1 to the Q3 of the distribution and

represents the IQR. A line across the box indicates the median. The whiskers are lines extending from Q1 and Q3 to end points that defined as the most extreme data points within Q1 − 1.5 × IQR and Q3 + 1.5 × IQR, respectively. *P* value was derived by two-sided permutation-based *t* test. **E** The correlation of FT3 and PA at blood routine level, the translucent bands around the regression line indicates the 95% confidence interval. The correlation coefficient was calculated by the pearson algorithm. *P* value was derived by the two-sided test. **F** The boxplots show the correlation of FT3 and lipid-related (*n* = 87), glucose-related (*n* = 12), and neuron-related (*n* = 14) biological processes, separately. The scatter size indicates the significance. The box ranges from the Q1 to the Q3 of the distribution and represents the IQR. A line across the box indicates the median. The whiskers are lines extending from Q1 and Q3 to end points that defined as the most extreme data points within Q1 − 1.5 × IQR and Q3 + 1.5 × IQR, respectively. *P* values were derived from the two-sided permutation-based *t* test. Source data are provided as a Source Data file.

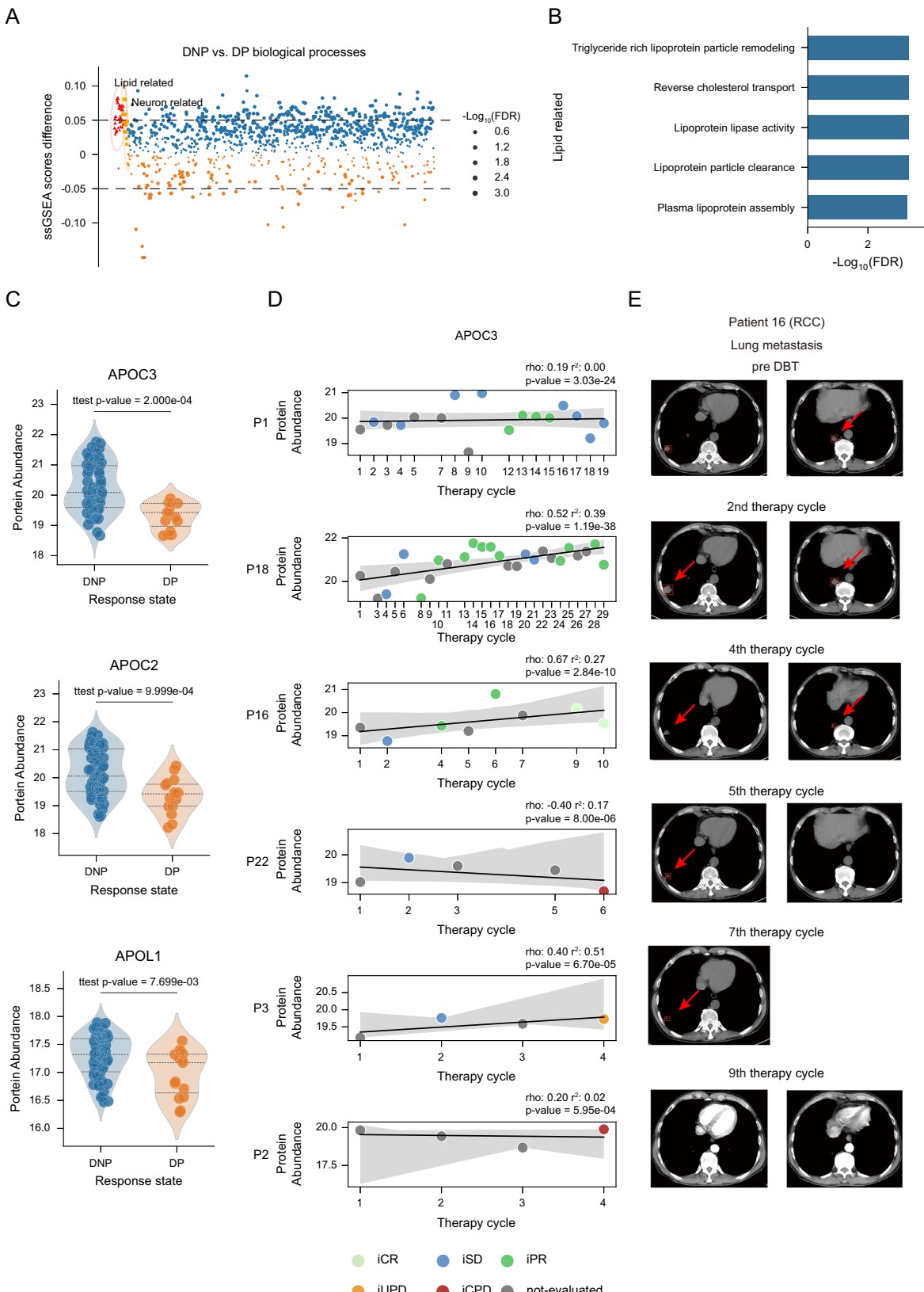

selected as the final model. After determining the best model for each of the feature combinations, in order to evaluate the model performance, we randomly split the cohort for 100 times to re-trained the model and re-evaluated the performance with more evaluation type including the area under the ROC curve (AUROC), balanced accuracy, F1, precision, recall, macro, and weighted evaluation metrics.

We found the two models with the APOC3 as feature and the integrated all features showed better performance than that in other models with the clinical indicators (Figs. 5B, S8B). We evaluated the model performance of the model with integrated all features, the AUROC, balanced accuracy, F1, recall, and precision were 0.87, 0.85, 0.79, 0.85, and 0.73. Furthermore, we found the LDH had the highest

**Fig. 4 | Cholesterol related APOC3 showed potential as biomarker in DBT. A** The dot plot shows the difference of the gene set ssGSEA scores between DNP and DP samples. The dot size indicates the FDR of the adjusted two-sided permutation-based *t* test *p* values. The top related biological processes including lipid-related and neuron-related were annotated as red and orange. **B** Bar plot depicting the significance of lipid-related biological processes in DNP samples. **C** The violin plots describe the level of apolipoproteins including APOC3, APOC2, and APOL1 between DNP and DP samples. *P* values were derived by two-sided permutation-based *t* test.

The line represents median, upper and lower quantiles, respectively. **D** The panel shows the time series-linear regression of APOC3 in 6 patients (P1, P16, P18 with all DNP samples, and P2, P3, P22 with DP samples). Dot color represents the evaluated response state. The translucent bands around the regression line indicates the 95% confidence interval. *P* values were derived by the two-sided F test. **E** The two tumor metastases radiology imaging of P16 with longitudinal samples. Source data are provided as a Source Data file.

importance, followed by the PA and APOC3 during the model decision (Fig. 5C and Supplementary Data 5) (*Methods*).

Encouraged by these results, we wondered whether the usage of biological biomarker could spread from DBT to the ICIs monotherapy, especially in anti-PD1 monotherapy. We collected the independent validation cohort of anti-PD1 monotherapy consisted of 54 longitudinal samples from 27 patients and implemented the same plasma proteome profiling pipeline as the DBT cohort (one sample not pass the quality control during the MaxLFQ quantification and excluded for the further validation) (Supplementary Data 5). For the five trained models, we calculated the confusion matrix (Figs. 5D, S8C) and then evaluated the AUROC, balanced accuracy, and F1 metrics on the independent validation cohort. These three metrics were 0.576, 0.672, and 0.421 for the model with APOC3 as feature; 0.681, 0.592, and 0.308 for the LDH; 0.983, 0.853, and 0.750 for the PA; 0.932, 0.882, and 0.700 for the integrated clinical features; and 0.961, 0.944, and 0.762 for the integrated all features. As shown in Fig. 5E, the model with integrated all features showed the better performance, followed by the model with integrated clinical features. Furthermore, we observed the APOC3 protein level of a patient with 6 samples points was gradually increased during the anti-PD1 monotherapy (Fig. S8D). These results indicated that PA, LDH, and APOC3 also have the potential to be predictive biomarkers for the response of anti-PD1 monotherapy.

In summary, we optimized plasma proteome profiling pipeline and provided the proteome dataset for the DBT cohort. By integrating the clinical and proteome data, we found the linkage of thyroid hormone indicators in clinical level and cholesterol processes in proteome level. Furthermore, we mined the key biomarker candidates PA, LDH, and APOC3. The proteome-clinical features-based machine learning model provides an accurate prediction on the DBT cohort. Notably, the model predictive ability could also spread to the anti-PD1 therapy cohort. Meanwhile, these clinical and proteomic biomarkers which explored by the longitudinal could also indicate the long-term response to the DBT for patients. In the end, this study extends our biological understanding of plasma proteins of DBT and generates hypotheses that may serve as the basis for future clinical trials toward the response of precision immunotherapy.

## Discussion

ICIs therapy changes patients' biological processes of the immunity and metabolism, and still much intrinsic biological characteristics need to be studied. The majority of research has focused on monotherapy with clinical, physiological or multi-omics data research[36], while there are few DBTs study. In this work, we aimed to describe global alteration of the plasma proteome in a systemic view and explore the potential biomarker of the DBT. We collected the longitudinal samples from patients who treated by the QL1706 and performed the advanced sample preparation, in-depth MS detection, MS data preprocessing and bioinformatic analysis, building a comprehensive proteome workflow[22]. This longitudinal study of DBT provides proteome fluctuations at plasma level, which facilitates biomarker exploration and biological mechanism speculation. In this study, we described the linkage of DBT response, thyroid hormone and cholesterol metabolism. Furthermore, we proposed PA, LDH, and APOC3 as potential response biomarkers for DBT.

LDH is a well-established biomarker in immunotherapy, especially in anti-PD1 monotherapy[13]. An elevated LDH has also been previously described to correlate negatively with OS in patients treated with ipilimumab and pembrolizumab[37–39]. As shown in Figure S4D, LDH level in plasma shows significantly positive correlation with neutrophil degranulation, which is consistent with the previous published study[40]. Thus, the decreased level of LDH in DNP samples was not surprising. However, we observed PA, rather than LDH, showed the most significant difference between DP samples and DNP samples, even in the pairwise level analysis in this QL1706 DBT cohort. Although the role of LDH in immunotherapy, especially the monotherapy is clear, it did not show the optimal predictive performance in the QL1706 DBT cohort.

The rash, hypothyroidism, pruritus, and hyperthyroidism were the four most common irAEs in the QL1706 phase I/Ib clinical trial. In this DBT cohort, the elevated level of PA indicates the activation of thyroid hormone[40] in DNP group of the DBT. Additionally, the thyroid hormone related clinical indicator FT3 and T3 provided the evidence. Hyperthyroidism is one of the most common irAEs in the anti-PD1 or anti-CTLA4 monotherapy, which is significantly associated with the prolong of overall survival and progression free survival[14]. The researchers detected the occurrence of hyperthyroidism might cause by acute destructive thyroiditis, suggesting the activated immune system and therefore associated with a better survival outcome[15]. In our cohort, the immune-related processes' scores were significantly correlated with PA (Supplementary Data 3). Notably, the lipid-related, especially the cholesterol-related biological processes also showed the positive associations with the PA. Correspondingly, it has been reported that the hyperthyroidism exhibits an enhanced excretion of cholesterol[41]. In summary, based on the plasma proteome data and the knowledge of thyroid hormone and immune, we illustrated the connections among thyroid hormone, cholesterol processes and DBT response, which contributes to the understanding of the role of hyperthyroidism during the DBT, as described in the phase I/Ib clinical trial.

In this cohort, we observed the enrichment of HDL-related processes (APOC2, APOC3, and APOL1) in DNP samples with elevated FT3, simultaneously, the reverse cholesterol transport process was also upregulated in DNPs (Fig. 4B). These results implied the increasement of cholesterol in blood, which is consistent with the published study[35]. Notably, it is reported that the inhibition of cholesterol metabolism and esterification would potentiate the efficiency of CD8 + T cell mediated antitumor response[42,43]. Based on the results, if interpreted the increasement of blood cholesterol as the results of the inefficiency of its metabolism, may represent the function of thyroid hormone of DNPs in the immune system activation. Clearly, the specific causal relationship between thyroid hormone, cholesterol, and immunity in the DBT needs to be further investigated.

The strengths of this study included a deeply phenotypic cohort with longitudinal samples, related blood routine, and radiology imaging, which allowed us evaluate the fluctuations of proteins and blood routines by setting the treatment time as the covariate. Notably, the radiology imaging provides a directly way to reflect the association between the clinical and the molecular alteration, hence intuitively validating the clinical value of biomarkers. In this study, the radiology imaging data confirmed the predictive potential of APOC3 in the DBT

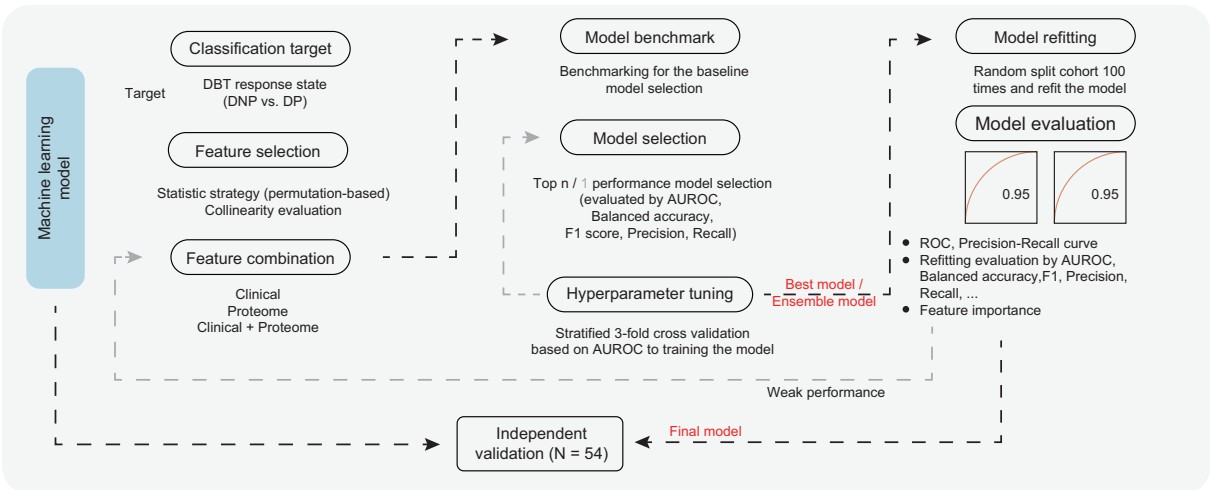

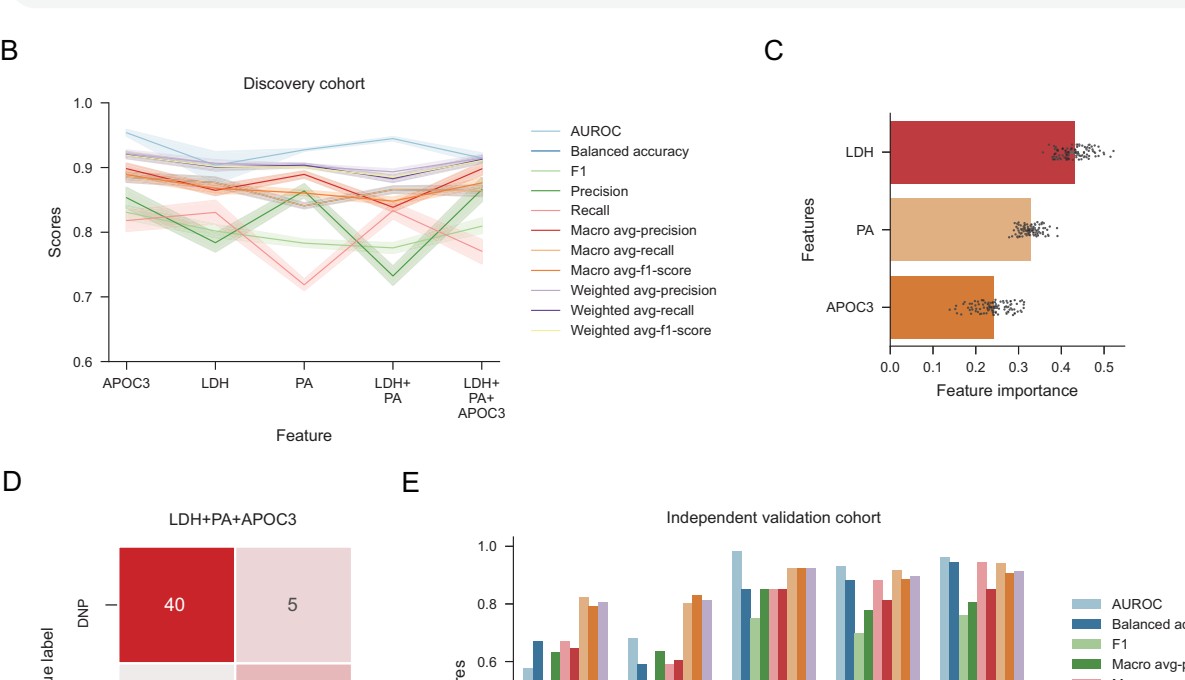

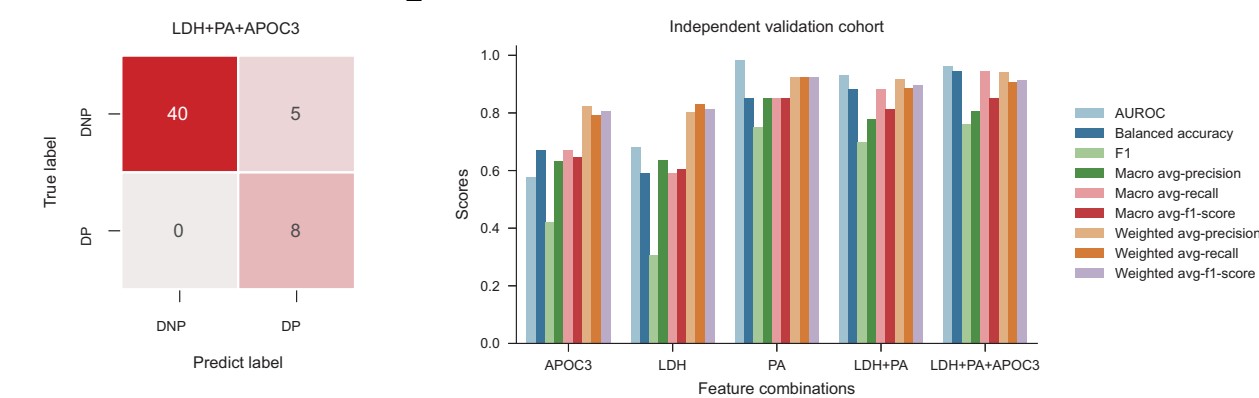

**Fig. 5 | Predicting response to DBT using a machine learning framework.**
**A** Schematic of the machine learning pipeline for the predictive model of the patient response to the DBT. The pipeline includes seven parts as feature selection, feature combination, model benchmarking, model selection, parameter tuning, model refitting, and model evaluation. **B** The different evaluation metrics including the area under the receiver operating characteristic curve (AUROC), accuracy, precision, recall, F1, macro and weighted scores for models with different feature combinations. The metrics were evaluated by the different models that were refitted on the 100 different cohort splits. The data was represented by the mean value and the color band indicates the 95% confidence interval. **C** The feature importance for the model with the integrated all features. The data was represented by the mean value and the error bar indicates the 95% confidence interval (repeats n = 100). **D** The confusion matrix of the models with different feature combination on the independent validation cohort. **E** The bar plot depicting the model performance on the independent validation cohort. The different evaluation metrics were used for the evaluation including AUROC, accuracy, precision, recall, F1, macro and weighted scores. Source data are provided as a Source Data file.

response. In addition, it is outperformed that the model with combining clinical and molecular features compared to that with only clinical features in the DBT response prediction. The high accuracy obtained in the independent validation cohort indicates that the

models are robust and may enable using the phenotype features to determine the therapy response in the future clinical trials. More generally, the multi levels of phenotype features and multiple validation strategy could be enrolled in the longitudinal cohort to investigate

the intrinsic characteristic of DBT and construct the more robust predictive model.

There are published clinical trials were proved the clinical benefits of DBT for the patients with advanced tumor. Exploring predictive biomarkers for the response state to DBT could benefit the patient in the clinical practice. Our longitudinal DBT cohort study was limited to 22 patients with 137 samples due to this cohort was collected from the phase I/Ib clinical trial. In the future, the larger-scale studies, including both more cases and more cancer types, will increase the power of the proteome and clinical biomarkers in indicating the response state of DBT. Hence, these results can serve as a basis for the future work which benefit for the clinical practice.

## Methods

### Patient samples of DBT cohort

The study was compliant with the ethical standards of Helsinki Declaration II and was approved by the institutional review board of Research Ethics Committees of the Affiliated Hospital of Hebei University (HDFY-LL-2020-157). Written informed consent was obtained from each patient before any study-specific investigation was conducted.

We have collected two cohort in this study including the discovery cohort (DBT cohort, $n = 22$) and the independent validation cohort (anti-PD1 cohort, $n = 27$). As for the DBT cohort, the samples were collected from the published phase Ib, open-label clinical trial cohort of the bifunctional PD1/CTLA4 dual blocker in patients with advanced solid tumors[11]. This clinical trial[11] is regulated and overseen by the National Medical Products Administration and the Ministry of Science and Technology. Compared with healthy donors, the dietary of the patients enrolled in DBT cohort was not specially planned. QL1706 was administered at the recommended phase 2 dose intravenously once every 3 weeks in patients[11]. Blood tests were done before the treatment and every three weeks, the radiology imaging was evaluated every six weeks. The radiology imaging was used for evaluating the tumor size and the response state.

Among the 22 patients enrolled in this DBT cohort, there are 6 patients with LC (lung cancer), 4 patients with CCA (cholangiocarcinoma), 3 patients with RCC (renal cell carcinoma), 3 patients with OVCA (ovarian cancer), 2 patients with CRCA (colorectal cancer), 2 patients with CESC (cervical squamous cell carcinoma), a patient with BLCA (bladder cancer), and a patient with UCEC (uterine corpus endometrial carcinoma). Additional clinical information such as gender, age, tumor node metastasis (TNM) stage, and routine blood test results, were listed in Supplementary Data 1.

### Proteomic workflow

**Plasma proteome samples preparation.** Plasma was separated by centrifugation at 16,000 g for 10 min within 30 min to remove insoluble solids and stored at −80 °C until proteomic analysis. The protein concentration of each sample was measured and recorded using the BCA method on a NanoDrop (Thermo Fisher Scientific) at 562 nm absorbance (Supplementary Data 1). 2 µL of plasma samples were mixed with 98 µL 50 mM ABC buffer, and protein inactivated at 95 °C for 3 min. The samples were cooled to room temperature. The plasma was digested by trypsin at an enzyme to protein mass ratio of 1:25 for 17 h at 37 °C. Then, 5 µL aqueous ammonia was added to each tube, vortexed to quench the digestion reaction, and the supernatant was subsequently dried using a 60 °C vacuum drier (SpeedVac, Eppendorf). Then, the peptides were dissolved in 100 µL 0.1% formic acid (FA), followed by vortexing for 3 min, and then sedimentation for 3 min (12,000 g). The supernatant was picked into new tube and then desalinated. Before desalination, the activation of pillars with 2 slides of 3 M C18 disk is required, and the lipid is as follows: 90 µL 100% acetonitrile (ACN) twice, 90 µL 50% and 80% ACN once in turn, and then 90 µL 50% ACN once. After pillar balance with 90 µL 0.1% FA twice,

the supernatant of the tubes was loading into the pillar twice, and decontamination with 90 µL 0.1% FA twice. Lastly, 90 µL elution buffer (0.1% FA in 50% ACN) was added into the pillar fir elution twice and only the effluent was collected for MS. Finally, the collected peptides were dried using a 60 °C vacuum drier.

**Nano-LC-MS/MS.** For the plasma proteome profiling samples, peptides were analyzed on a Q Exactive HF-X Hybrid Quadrupole-Orbitrap Mass Spectrometer (Thermo Fisher Scientific) coupled with a high-performance liquid chromatography system (EASY nLC1200, Thermo Fisher Scientific). Dried peptide samples re-dissolved in 50 µL Solvent A (0.1% formic acid in water). The peptide concentration of each sample was measured and recorded using a NanoDrop (Thermo Fisher Scientific) at 280 nm absorbance, and ultimately a standard loading amount of 200 ng peptide were loaded on a 75 µm-inner-diameter column with a length of 9 cm (1.9 µm ReproSil-Pur C18-AQ beads, Dr Maisch GmbH) over a 15 min gradient (Solvent A: 0.1% Formic acid in water; Solvent B: 0.1% Formic acid in 80% ACN) at a constant flow rate of 600 nL/min (0 min, 6% B; 0−6 min, 6−30% B; 6−8.20 min, 30−50% B; 8.20−9.2 min, 50−95% B; 9.2−12.3 min, 95% B; 12.3−13.3 min, 3% B; 13.3−15 min, 3% B). Eluted peptides were ionized at 2 kV and introduced into the mass spectrometer. Mass spectrometry was performed in data-independent acquisition mode (DIA). For the MS1 Spectra full scan, ions with m/z ranging from 300 to 1400 were acquired by an Orbitrap mass analyzer at a high resolution of 30,000. The automatic gain control (AGC) target value was set to 3E06. The maximal ion injection time was 20 ms. Then, the DIA segments were required at 15 K resolution with an AGC target of 1e6. The default charge state for the MS2 was set to 2.

**MS database searching.** The MS database searching included two aspects as the hybrid library construction and DIA data analyzing. All data were processed using Firmiana[44].

**Hybrid library construction.** Owing to the plasma protein were secreted from the tissue, the tissue proteome was the ideal library for the DIA plasma proteome. In addition, this DBT cohort was collected from a phase I/Ib clinical trial and 8 tumor types were enrolled in this cohort. Hence, we collected a total 327 DDA raw files derived from the different tumor tissues and performed the recommended library construction pipeline. We processed all 327 of the newly acquired and collated proteomic datasets to generate the protein spectral libraries. The raw MS files underwent conversion to the mzML file format utilizing MSConvert software. For the construction of consolidated spectral libraries, we engaged the FragPipe computational platform (version 12.1), equipped with MSFragger (version 2.2)[45], Philosopher (version 2.0.0)[46], and Python (version 3.6.7). The converted DDA mzML files, throughout the spectral library construction, were cohesively processed.

Peptide identification was executed from MS/MS spectra using FragPipe integrated with the MSFragger search engine. This was matched against the UniProt human protein database (reviewed sequences only; as updated on 2019.12.17, housing 20,406 entries), which also encompassed reversed protein sequences appended as decoys for ensuing false discovery rate (FDR) calculations. Technical specifications included both precursor and initial fragment mass tolerances set at 20 ppm, enabling spectrum deisotoping, mass calibration, and parameter optimization. We established enzyme specificity to 'trypsin' and permitted up to 2 missed trypsin cleavages. Configurations for peptide length ranged between 7 and 25, while peptide mass was set from 500 to 5000 Da. Only precursor ion score charges of +2, +3, and +4 were considered. Cysteine carbamidomethylation was fixed as a modification, while variable modifications entailed N-acetylation and methionine oxidation. A maximum of three variable modifications per peptide was allowed.

Subsequent to the search, MSFragger search results (in pepXML format) were processed using the Philosopher toolkit. PeptideProphet (run with the high–mass accuracy binning and semi-parametric mixture modeling options) was employed to compute the posterior probability of correct identification for each peptide-to-spectrum match (PSM). The emergent output files from PeptideProphet were processed together using ProteinProphet to perform protein inference (assemble peptides into proteins) and synthesize a unified protXML file containing high-confidence protein groups. The combined ProteinProphet file was further processed using the Philosopher Filter command, which characterized each identified peptide as a unique peptide to a particular protein (or protein group containing indistinguishable proteins) or assigned it as a razor peptide to a single protein (protein group) that had the most peptide evidence. Both unique and razor peptides were used for subsequent analysis. The data were filtered to 1% protein-level FDR using the picked FDR strategy. The peptide-, PSM-, and ion-level reports were then generated and filtered using the 2D FDR approach (i.e., 1% protein FDR plus 1% PSM/ion/peptide-level FDR for each corresponding PSM.tsv, ion.tsv, and peptide.tsv file). The resulting hybrid spectral library contained 215,529 precursors, representing 15,612 proteins.

**DIA data analyzing.** DIA data was analyzed using DIANN (v1.7.0) against the hybrid library[47]. The DIANN search included the following settings: Precursor FDR: 1%, Log lev: 1, Mass accuracy: 20 ppm, MS1 accuracy: 10 ppm, Scan window: 30, Implicit protein group: genes, Quantification strategy: robust LC (high accuracy). Label-free protein quantifications were determined using an intensity-based, label-free approach incorporating delayed normalization and maximal peptide ratio extraction (MaxLFQ) approach[28]. Peak area values were computed as components of their respective proteins. The 'iq' R library was used for the MaxLFQ quantification[48]. There were 137 samples in DBT cohort and 53 samples in the independent validation cohort were quantified, finally.

**Proteome data preprocess**

**Mass spectrometry platform QC.** As for investigating the quantitative reproducibility of our workflow, we randomly selected 10 samples from the DBT cohort. Each sample was detected in consecutive replicates 5 times using the same MS method as for the actual samples of the DBT cohort. We calculated the coefficient of variance and the pearson correlation for each of the sample among the 5 repeats.

As for evaluating the quality control of the MS performance during the plasma sample detection, we mixed all of samples in this study into a plasma pool as QC standards. The QC standards were analyzed using the same method and conditions as the DBT cohort. The coefficient of variance was calculated to evaluate the stability of the mass spectrometry platform.

**Data preprocessing.** Considering the balance between the confidence of protein identification and sample heterogeneity, the analysis in this study focused on the proteins identified in >70% of samples. The missing value was served as NaN for the downstream analysis.

**Batch effect evaluation.** The batch effect was evaluated for plasma proteome data using principal component regression analysis. Specifically, we set the two evaluated metrics to evaluate the batch effects as (1) count the number of the significant correlated principal components; (2) calculated the batch related information (brinfo) as the following formula:

$$\text{brinfo} = \sum_{i=1}^{n} \text{abs}\left(\text{rho}_i\right) \times \text{eigenvalue}_i \tag{1}$$

For which the $\text{rho}_i$ is the spearman correlation between principal component and the potential batch, $n$ is the count of the principal components (PC). The brinfo represented the fraction of the proteomic data information that correlated with the specific batch.

**Quantification and statistical analysis**

**Protein clusters.** We grouped the DEPs among healthy control group, pre DBT group and 1st DBT cycle group into four clusters (HLH, LMH, LHL, HML) by the relative z-score value among three sample groups. In detail, in the HLH protein cluster, the proteins were upregulated in the healthy control and 1st DBT group than the pre DBT group. The levels of proteins in LMH are consistently elevated in the three sample groups. The levels of proteins in LHL only upregulated in the pre DBT group. The proteins in HML group were consistently decreased among three sample groups.

**Gene set Score for Single Sample.** We calculated the normalized enrichment score of each sample based on four class of gene sets: GOBP, KEGG, Hallmark, and Reactome gene sets. We utilized R package GSVA[49] with following parameters: min.sz = 3, max.sz = 200 and other parameters were used default.

**Time-series linear regression.** To determine the core biomarker in the longitudinal samples, we used the individuals in the study and tested the linear regression model with the treatment points as the covariable. The type II sum of squares was calculated using the OLS function of the statsmodel packages in python.

**Machine learning models.** Machine learning was conducted to identify the response to DBT. The graphical machine learning model construction pipeline is illustrated in Fig. 5A and includes seven parts: feature selection, feature combination, model benchmark, model selection, hyperparameter tuning, model refitting, and model evaluation. The machine learning pipeline was built in Python (version 3.9.15) using the following libraries: scikit-learn (version 1.2.1), numpy (version 1.26.3), scipy (version 1.12.0), and pandas (version 1.5.3).

**Feature selection.** The permutation-based $t$ test were used for determined the significantly different clinical indicators and proteins as described in Figs. 3 and 4. The features that the correlation coefficient large than 0.7 were excluded for the downstream model construction.

**Feature combination.** We designed the seven different feature combinations as (1) PA, (2) LDH, (3) APOC3, (4) integrated clinical features (PA and LDH), (5) integrated all features (PA, LDH, and APOC3). We carried out the same machine learning construction pipeline to provide a comparable benchmarking result.

**Model benchmarking and model selection.** To choose machine learning with good performance, we compared 21 different models and used the average value of the AUROC, balanced accuracy, F1, precision, and recall to evaluate the model performance. The top 10 models with higher average value were selected for the downstream hyperparameter tuning.

**Hyperparameter tuning.** To balance the calculation burden and the model accuracy, we used the randomized search cross validation strategy to search the optimal hyperparameter, setting the roc_auc as the scoring function and 3-fold cross validation.

**Model refitting and Model evaluation.** In order to evaluate the model performance, we randomly re-split cohort and refit the model for 100 times, and evaluate the AUROC, balanced accuracy, F1, precision, and

recall on the whole discovery cohort. The feature importance was calculated as the average of the 100 times refitting.

**Predictor validation.** We used the outer independent validation cohort. The independent cohort was the collected anti-PD1 ICIs treatment cohort with longitudinal samples. The models were applied on the validation cohort and the metrics of the confusion matrix was evaluated.

**Statistics and reproducibility.** Standard statistical tests were used to analyze the clinical data, including but not limited to Student's $t$ test, Chi square test, Pearson's correlation test, Spearman's correlation test, F test, and Kruskal test. Unless otherwise specified, all statistical tests were two-sided. P values less than 0.05 were considered as significantly different. To account for multiple testing, the P values were adjusted using the Benjamini-Hochberg FDR correction. All the analyses of clinical data were performed in R (v4.0.0) and python (v3.9.15). No statistical method was used to predetermine sample size. One sample in the independent validation cohort was excluded for the downstream analysis due to not pass the quality control during the proteome quantification. For ESI-LC-MS/MS analysis, the longitudinal samples from different patient and different therapy cycle were detected with a random order to exclude the bias effects of the mass spectrometry. As for the machine learning model construction, the samples were randomly divided into train and test cohort. The investigators who measured protein expression were blinded to patient information.

#### Reporting summary
Further information on research design is available in the Nature Portfolio Reporting Summary linked to this article.

## Data availability
The proteome raw datasets have been deposited to the ProteomeXchange Consortium (dataset identifier: PXD039260) via the iProX partner repository[50] (https://www.iprox.cn/) under Project ID: IPX0005695001 (https://www.iprox.cn/page/PDV0141.html). In detail, the data of the discovery cohort and independent validation cohort were uploaded, separately: Annotation file: the annotation of sample information including sample name, therapy cycle, etc.; MS raw data: the MS raw data; FASTA file: the FASTA file used for the MS data processing; DIA-NN output file: the merged DIA-NN standard output file; Proteome expression matrix: the quantification matrix data. The more detailed description could be found in the README file under the above link. The remaining data are available within the Article, Supplementary Information. Source data are provided with this paper.

## Code availability
All sources of code for computational analyses were derived from publicly available websites and previous publications, and are cited in the corresponding Methods sections. The code used for this study was deposited at GitHub (https://github.com/Jiacheng-Lyu/DBT-plasma-proteome). The code was also deposited at Zenodo (https://doi.org/10.5281/zenodo.10824474).

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

## Acknowledgements

This work is supported by National Key Research and Development Program of China (2022YFA1303200 [C.D.], and 2022YFA1303201 [C.D.]), National Natural Science Foundation of China (32330062 [C.D.], and 31972933 [C.D.]), sponsored by Program of Shanghai Academic/Technology Research Leader (22XD1420100 [C.D.]), the Major Project of Special Development Funds of Zhangjiang National Independent Innovation Demonstration Zone (ZJ2019-ZD-004 [C.D.]), Shanghai Municipal Science and Technology Major Project (2017SHZDZX01 [C.D.]), the Fudan Original Research Personalized Support Project [C.D.], Cultivation Project of Precision Medicine Joint Fund of Hebei Natural Science Foundation (H2022201044), 2021 Government-funded clinical talents Training Project: Small cell lung Cancer Basic and Transformation Innovation Research team (361007), and Innovative team for precise care and rehabilitation of patients with cancer (No. IT2023C07). This work is supported by the Human Phenome Data Center of Fudan university.

## Author contributions

C.D., Y.C.J., J.C.L., L.B., and Y.M.L. conceived, designed and organized the study. Y.M.L., X.F.W., H.Y., Z.Z.S., Z.Y.W., Y.H.S., L.L.R., Y.L. A.M.Z. were responsible for sample and clinical information collection. J.C.L., L.B., T.J., and Z.Y.X. contributed to sample preparation. J.C.L. and L.B. analysis the data. J.C.L., L.B., Y.M.L, and X.F.W. interpreted the results and drafted the manuscript. All authors provided critical feedback and helped shape the research, analysis, and manuscript.

## Competing interests

The authors declare no competing interests.
