## [Peer Review File · Nature Communications]

Plasma proteome profiling reveals dynamic of cholesterol marker after dual blocker therapyREVIEWERS' COMMENTS

Reviewer #1 (Remarks to the Author): expertise in assessing clinical response to dual blocker therapy

The authors of this manuscript are to be congratulated on a well written and thought-out study, with rigorous methodology and analyses.

Although the small sample size is an obvious limitation to further interpretations, these results can serve as a basis for future work which could be clinically impactful.

I would recommend reviewing the introduction for English fluency, as that does not flow that well at present, and detracts from the excellent findings in the rest of the manuscript.

Reviewer #2 (Remarks to the Author): expert in proteomics analysis of treatment response

Lyu et al, presents their work on the 'Plasma proteome profiling reveals dynamic of cholesterol marker after PD1/CTLA4 dual blocker therapy'. The work describes the need for a biomarker to monitor clinical response to bispecific antibody therapy. For this the authors have used longitudinal plasma samples collected from the patient who underwent the DB therapy and reports lipid carrier APOC3 as a putative biomarker for response in a small cohort of samples. APOC3 secreted by liver, inhibits lipoprotein lipase and its increase is known to indicate the vLDL and cholesterol levels. This study did not rule out whether this change is due to hospitalized diet as the comparison is with healthy donors (whose diet plan is not in the method section). Further the longitudinal APOC3 response is not a definite marker and the data is not impressive as well despite of a validation cohort while it is also not clear which tumor samples were studied. The ML prediction model reports an accuracy of >0.8, which again is an indication of lack of predictive power due to small sample size. Importantly, this study did not follow the proteomic community rigor and reporting. Here are my comments on this manuscript

1. The introduction is clear and concise but there are grammatical errors throughout the manuscript and makes it difficult to understand at several statements.
2. They did not mention throughout the manuscript in which clinical tumor types were administered on DBT. Line 132, among 64 samples, but which disease ? and what response are you discussing here ? Neither Figure 1 nor its legends say anything about the tumor. Actually what tumor is this cohort ?
3. I suggest the sample size be increased to atleast >50 for both the cohort and reanalysis of the ML data.
4. The naming of the groups is not very clear, especially the "before DBT group" – I suggest to change it to "Pre DBT group".
5. The Proteomics data repository does not have any annotations on which files are pre DBT, control or 1st DBT. I request to upload the following data 1. Annotation file, 2. FASTA file used for the analysis, 2. DIA-NN output. As a reviewer we tried to evaluate the data on and we find it stressful that the authors did not follow any proteomic community standards.
6. Line 510-511, Quantification of peptides in DIA-NN is not the average of chromatographic fragment ion peak areas but top 6 ions.
7. Calculation of Protein Quantity by Fraction of total (FOT) is not the most used way. Was the quantification done using only the unique peptides or using unique and razor peptides?
8. Calculating MaxLFQ may help reduce the CVs even further.
9. The result file in the repository has only one column with FOT, were all time points merged? This

kind of reporting is see in a black box.

10. It is not very clear what protein abundance value the authors use. Is it fraction of total value calculate for each sample? Or Is it iBAQ value?

11. Lines 220-221, Grouping of iPR and iSD could be changed to "disease non-progressive" and not, "non-disease progression".

12. Line 197, it is good to indicate fold changes instead of p values for particular proteins.

13. Line 177-179, The authors used p-Value ≤ 0.05 as a criteria for significance, it is highly recommended to apply an additional FDR control.

14. Line 181, I suggest to provide the z-score values as an additional supplementary table.

15. Figure 2 A, X and Y axis titles should be included in the figure instead of explaining in the legend separately. As it makes it difficult to understand if the ratios are before DBT/1st DBT or 1st DBT/before DBT.

16. In Supplementary figure 1, the authors have shown the correlation and CV values for the QC samples only, they could also provide the same for the actual samples.

17. Line 356, No proteomics data for the validation cohort has been uploaded to the repository. I recommend to upload this data as well with the additional data mentioned in point 3.

18. Supplementary 5 and 6 does not serve the purpose for this study. What is the point of having this scan while the study is not to assess the DBT response outcome

Reviewer #3 (Remarks to the Author): expert in proteomics model development

The team sought to examine the biomarkers for dual blocker therapy in a proteomics study. The team is known for the proteomics platform and has built track records on proteomics studies.

1. Although the study reports interesting findings on proteomic profiles after DBT treatment, the clinical relevance of the findings should be strengthened. If the biomarkers for DBT response can be used as "intermediate" biomarkers for the survival of patients or other long-term outcome after DBT response, the clinical significance may be better justified.

2. Following the last comment, if the authors can better justify the clinical application of their findings, how frequently would physicians perform proteomics study to monitor the response? How would physicians read the interpret the findings?

RESPONSE TO REVIEWERS' COMMENTS:

Reviewer #1 (Remarks to the Author): expertise in assessing clinical response to dual blocker therapy

Q1. The authors of this manuscript are to be congratulated on a well written and thought-out study, with rigorous methodology and analyses. Although the small sample size is an obvious limitation to further interpretations, these results can serve as a basis for future work which could be clinically impactful.

Response:

We sincerely thank the reviewer for the positive comments about the methodology and analyses of this manuscript. Herein, in order to explore the biomarker that can indicate the response state during the dual blocker therapy (DBT) for patients, we collected the first longitudinal DBT cohort in the worldwide consisting of a total of 86 proteome profiling of plasma samples (we further expanded the samples from 86 to 103 in the revised version) from 22 patients with 8 different tumors. The main findings of this study included four parts as (1) the PI3K-AKT signaling was downregulated, while immune response and cholesterol metabolism were upregulated after the first DBT cycle, and the cholesterol metabolism was activated in the non-disease progression (NDP) group (Since the informal nomenclature, it was changed to the **disease non-progressive (DNP)** group in this revision according to the suggestion of Reviewer #2) during the whole DBT cycle; (2) the clinical indicator prealbumin (PA) and free triiodothyronine (FT3) showed significantly positive association with the cholesterol metabolism; (3) APOC3, participating in high-density lipoprotein partial remodeling, was a biomarker candidate that can reflect the response during the DBT; (4) the machine learning model with PA, LDH, and APOC3 as feature showed the good performance for distinguishing the DNP group from the disease progressive (DP) group.

These main findings needed to be supported by the large cohort. Herein, owing to the purpose of this cohort study and the rigorous cohort design principal, the sample enrollment was indeed restricted. We admitted the small sample size of the discovery cohort (DBT cohort) is an obvious limitation. In order to avoid the negative impact of cohort size on the mentioned above main study

results to a certain extent, **we continued to collect samples from patients who were enrolled in the discovery cohort and still received DBT**. Meanwhile, we also **expanded the sample size of the independent validation cohort** to validate the machine learning performance. Ultimately, the discovery cohort contained **103** samples (**86** samples in the previous version) and the independent validation cohort consisted of **54** samples (**43** samples in the previous version). We re-analyzed the data and found the main results were robust, the detailed information as follows.

As for the main result of the clinical indicator prealbumin (PA) and free triiodothyronine (FT3) showed significantly positive association with the cholesterol metabolism

We compared the clinical indicators between DNP and DP group (corresponding to the **Figure 3** in the previous version of the manuscript). **Figure RL 1A, B, and C** were the results derived from the cohort in the previous version of the manuscript and **Figure RL 1D, E, and F** were the results derived from the cohort in the revised version of the manuscript. As shown in **Figure RL 1D, E**, the lactate dehydrogenase (LDH) ($p\text{-value} = 0.023$), alkaline phosphatase (ALP) ($p\text{-value} = 8.01E-3$), and white blood count (WBC) ($p\text{-value} = 5.58E-3$) were significantly upregulated in DP group; and hemoglobin (HGB) ($p\text{-value} = 8.25E-6$), red blood cell (RBC) ($p\text{-value} = 4.94E-5$), free triiodothyronine (FT3) ($p\text{-value} = 0.017$), albumin globulin ratio (A/G) ($p\text{-value} = 5.00E-5$), prealbumin (PA, also known as thyroxine-binding prealbumin) level ($p\text{-value} = 1.18E-3$), and Creatine Kinase-MB (CK-MB) ($p\text{-value} = 1.49E-4$) were significantly downregulated in DP group. Furthermore, there was a significantly positive spearman's correlation between PA and FT3 ($\rho = 0.72, p\text{-value} = 4.00E-3$) (**Figure RL 1F**) as the result shown in **Figure RL 1C** ($\rho = 0.70, p\text{-value} = 0.012$), which indicated attenuation of thyroid hormone synthesis in DP samples. We have updated these results to **Figure 3** and page 9 **line from 229 to 261** of the manuscript in the revision.

Figure RL 1A. Heatmap depicting the different level of blood routines in the cohort of the previous version. Values were transformed by z-score. **B.** The boxplot describes the level of free triiodothyronine (FT3) and the prealbumin (PA) between non-disease progression (DNP) and disease progression (DP) samples at blood routine level in the cohort of the previous version. *P*-value was derived by t-test. **C.** The correlation of FT3 and PA at blood routine level in the cohort of the previous version. **D.** Same as **Figure RL 1A**, but for the cohort in the revised version. **E.** Same as **Figure RL 1B**, but for the cohort in the revised version. **F.** Same as **Figure RL 1C**, but for the cohort in the revision.

As for the APOC3 could be served as the candidate biomarker that reflect the response during the DBT

We compared the expression of key proteins which participated in the lipoprotein-related biological process and the linear relations with the therapy cycle (same as the **Figure 4** in the previous version of manuscript). As shown in **Figure RL 1G** and **H**, the APOC3 (*p*-value = 1.42E-3 in the revised version vs. *p*-value = 7.02E-3 in the previous revised version) and APOC4 (*p*-value = 0.043 in the revised version vs. *p*-value = 0.025 in the previous revised version) showed increased level in DNP group than that in DP group, and the APOL1 showed decreased level in DNP group than that in DP group (*p*-value = 5.82E-3 in the revised version vs. *p*-value = 4.28E-3 in the previous

revised version). The APOA4 showed significantly increased level in DNP group in the previous version (p -value = 0.037), but did not show significant difference in the revised version (p -value = 0.14), which indicated the non-robustness of APOA4 for indicating the response state during the DBT cycle. This protein was not the feature in the ML model. Therefore, the performance of the ML model was not affected.

Figure RL 1G. The boxplot describes the expression level of APOC3, APOC4, APOA4, and APOL1 in the cohort of the previous version. P -value was derived by t-test. **H.** Same as **Figure RL 1G**, but for the cohort in the revised version.

Furthermore, we also re-evaluated the linear relation between the expression of these lipoproteins and the DBT cycle for six patients, three of whom ultimately responded to the DBT (P1, P16, and P18) whereas other patients ultimately not responded to the DBT (P2, P3, and P22). As shown in **Figure RL 1I**, in the previous version, comparing with the other three lipoproteins (APOC4, APOA4, and APOL1), the protein APOC3 showed the best performance for indicating the response state during DBT due to the higher correlation and the best linear model fitness in DNP group, and the reverse tendency between DNP and DP groups. In the revision (**Figure RL 1J**), for the DNP group, APOC3 and APOC4 showed better performance than APOA4 because of the APOA4 in P1 showed lower spearman's rho ($\rho = 0.15$). In addition, the APOC4 showed precise

fitting and positive correlation with DBT cycle in the P3 who ultimately not responded to the DBT, which indicated the risk of false discovery for the prediction. While the APOC3 did not show significantly positive association with DBT in patients who ultimately did not respond to the DBT. Hence, the APOC3 was also the best biomarker candidate for predicting the response state during the DBT. Based on these results, we reconstructed the machine learning model for the cohort of the revision. We have updated these results to **Figure 4** and page 11 **line from 285 to 312** of the manuscript in the revision.

Figure RL 11. The panel shows the time series-linear regression of APOC3, APOC4, APOA4, and APOL1 in 6 patients (P1, P18, P16 with all DNP samples, and P22, P3, P2 with DP samples). Dot color represents the evaluated response state. **J.** Same as **Figure RL 11**, but for the cohort in the revised version.

As for the performance of machine learning model with LDH, PA, and APOC3 as features to distinguish DNP and DP group.

We enrolled an additional 27 samples for the discovery cohort and an additional 11 samples for the independent validation cohort. Finally, the discovery cohort contained 103 samples and the independent validation cohort contained 54 samples. By performing the same ML pipeline as mentioned in the previous version of manuscript, we reconstructed the ML model to distinguish the DNP and DP group with three features including LDH, PA, and APOC3 for the samples (62 samples)

which have the response states (**Figure RL 1K**). As shown in **Figure RL 1L, 1M**, the confusion matrix and receiver operating characteristic curve (ROC) of train, test, and independent validation cohort showed the good performance of the classifier model. The area under ROC (AUROC) for the three cohort were 0.96, 0.96, and 0.96. Furthermore, to evaluate the model performance, we randomly split the cohort for 100 times to re-trained the model and re-evaluated the performance with more evaluation type including AUROC, accuracy, precision, recall, and F1 score. In detail, the train cohort had average scores 0.96, 0.93, 0.94, 0.93, and 0.93 with standard derivation of 0.026, 0.028, 0.029, 0.028, and 0.030, test cohort had scores 0.76, 0.81, 0.81, 0.81, and 0.79 with standard derivation of 0.13, 0.06, 0.086, 0.060, and 0.074, the independent validation cohort had scores 0.90, 0.90, 0.91, 0.90, and 0.90 with standard derivation of 0.047, 0.052, 0.031, 0.052, and 0.044 (**Figure RL 1N**). The **Figure RL 1O** showed the probability of predicted samples during the classifier decision. The reconstructed model displayed the better performance than that in the previous version for classifying the DP group from DNP group, which further indicated the features including LDH, PA, and APOC3 had the power in the therapy response prediction task. We have updated these results to **Figure 5** and page 13 **line from 347 to 370** of the manuscript in the revision.

Above all the analysis, we found the main results were unchanged after adding more longitudinal samples, which indicated the important role of apolipoprotein, thyroid hormone during the DBT. Notably, enrolling more patients could help validate the findings and we believed these results can serve as a basis for the future work which benefit for the clinical practice.

Figure RL 1K. Schematic of the machine learning framework including four parts as Data collection, Feature selection, Cross validation (CV), and Test cohort validation. **L.** The confusion matrix of train, test, and validation cohort of the machine learning model. **M.** The receiver operating characteristic curve (ROC) of train, test, and validation cohort of the machine learning model. **N.** The evaluated scores for 100 repeated trained model. **O.** The bar charts depicting the machine learning model's predicted probabilities for samples in the discovery and validation cohorts.

Q2. I would recommend reviewing the introduction for English fluency, as that does not flow that well at present, and detracts from the excellent findings in the rest of the manuscript.

Response:

We thank the reviewer for the careful read and valuable suggestions to present the statement of our manuscript clearly and concisely. According to reviewers' constructive comments, firstly, we changed the ambiguous description and modified the imprecise conclusions. Then, after compiling all the materials, we adjusted arrangement of all kinds of supporting materials, including figures, tables, et al., according to our story line to highlight the main results and conclusion of the article. The revised manuscript was also edited by professional native speakers. All changes were highlighted with red text in the revised manuscript.

Reviewer #2 (Remarks to the Author): expert in proteomics analysis of treatment response

Lyu et al, presents their work on the ‘Plasma proteome profiling reveals dynamic of cholesterol marker after PD1/CTLA4 dual blocker therapy’. The work describes the need for a biomarker to monitor clinical response to bispecific antibody therapy. For this the authors have used longitudinal plasma samples collected from the patient who underwent the DB therapy and reports lipid carrier APOC3 as a putative biomarker for response in a small cohort of samples. APOC3 secreted by liver, inhibits lipoprotein lipase and its increase is known to indicate the VLDL and cholesterol levels.

Response:

We sincerely thank the reviewer for the careful reading and the summary of this manuscript.

Q3. This study did not rule out whether this change is due to hospitalized diet as the comparison is with healthy donors (whose diet plan is not in the method section). Further the longitudinal APOC3 response is not a definite marker and the data is not impressive as well despite of a validation cohort while it is also not clear which tumor samples were studied. The ML prediction model reports an accuracy of >0.8 , which again is an indication of lack of predictive power due to small sample size. Importantly, this study did not follow the proteomic community rigor and reporting. Here are my comments on this manuscript.

Response:

Thanks for the insightful comments and suggestion. In order to clearly respond to the comments, we split the comments into four parts, (1) the diet plan of patients, (2) the tumor samples composition, (3) the ML model construction, and (4) the proteomic reporting. We respond to the above four points one by one as follows.

As for the diet plan of patients, the phase I/Ib clinical trial was conducted to investigate the safety, tolerability, pharmacodynamics and preliminary efficacy of QL1706 in patients with advanced solid tumors. This clinical trial research has been published on *Journal of Hematology & Oncology* titled “First-in-human phase I/Ib study of QL1706 (PSB205), a bifunctional PD1/CTLA4 dual blocker, in patients with advanced solid tumors”. This clinical trial is regulated and overseen by the National Medical Products Administration and the Ministry of Science and Technology. Meanwhile, in order to explore the biomarker that could indicate the response state to the DBT, we

collected the longitudinal samples from the QL1706 DBT cohort. The patients in this study were compliant with the ethical standards of Helsinki Declaration and was approved by the institutional review board. We clarify that this plasma proteome analysis (this interim analysis) was prospectively registered (HDFY-LL-2020-157). **The patients enrolled in this cohort were collected from this clinical trial. Compared with healthy donors, their dietary was not specially planned.** In addition, the median of body mass index (BMI) of the cohort was 23.53, which was fall into the normal range (18.5 to 24.9). We compared the BMI between the disease non-progressive (DNP) and disease progressive (DP) and observed there was no significantly difference between them (**Figure RL 2A**, $p\text{-value} > 0.05$). We have added the description of diet for patients in the **Methods** part page 18 from **line 507 to 508** of the revision.

Figure RL 2A. The box plot depicting the BMI between DNP and DP samples.

As for the tumor samples composition, we apologized for the lack of detailed description for this cohort in the previous version of manuscript. We have responded this comment and updated this information in **Results** part page 5 from **line 114 to 118** and **Methods** Part page 19 from **line 511 to 517** of the revised manuscript. Please see the response of R2-Q5 for more details.

As for the ML model construction, we admitted the small sample size of the discovery cohort (DBT cohort) could affect the predictive power. To resolve this problem, **we continued to collect samples from patients who were enrolled in the discovery cohort and still receiving DBT.** Meanwhile, we also **expanded the sample size of the independent validation cohort** to validate the machine learning performance. Finally, the discovery cohort contained 103 samples (86 samples in the previous version) and the independent validation cohort contained 54 samples (43 samples in the previous version). We used the samples (62 samples) that have both the response state and LDH,

PA, and APOC3 expression level to reconstruct the ML model. The ML model showed the good performance manifested by the accuracy was 0.90 on the discovery cohort and 0.93 on the independent validation cohort. Please see the response of R2-Q6 for more details.

As for the proteomic reporting, we apologized for not following the proteomic community rigor and reporting. The proteome matrix and clinical information were uploaded in the **Supplementary Table 1** in the previous version of submission. For the easily and friendly availability of the proteomic information for the discovery and independent validation cohort, we have updated the MS raw data, sample annotation file, DIA-NN output, and related files to the iProX in this revision. Please see the response of R2-Q8 for more details.

Finally, we sincerely thank again for the reviewer's comments.

Q4. The introduction is clear and concise but there are grammatical errors throughout the manuscript and makes it difficult to understand at several statements.

Response:

Thanks for the comment. We apologized for the grammatical errors and the lack of fluency throughout the manuscript that made it difficult to understand at several statements. We have carefully checked the manuscript to correct grammatical and spelling errors. The revised manuscript was also edited by professional native speakers. All changes were highlighted with red text in the revised manuscript.

Q5. They did not mention throughout the manuscript in which clinical tumor types were administrated on DBT. Line 132, among 64 samples, but which disease? and what response are you discussing here? Neither Figure 1 nor its legends say anything about the tumor. Actually, what tumor is this cohort?

Response:

We apologized for the unclear description of this DBT cohort. Thanks again for pointing it out.

As for the tumor types in the DBT cohort, among the 22 patients enrolled in this cohort,

there are 6 patients with LC (lung cancer), 4 patients with CCA (cholangiocarcinoma), 3 patients with RCC (renal cell carcinoma), 3 patients with OVCA (ovarian cancer), 2 patients with CRCA (colorectal cancer), 2 patients with CESC (cervical squamous cell carcinoma), a patient with BLCA (bladder cancer), and a patient with UCEC (uterine corpus endometrial carcinoma). This information was mentioned in the **Supplementary Table 1** in the previous version of submission. To describe this directly and clearly, we have responded this comment and updated this information in **Results** part page 5 from **line 114 to 118** and **Methods** Part page 19 **from line 511 to 517** of the revised manuscript.

The DBT was an improved strategy for the monotherapy that can increase the immune response rates and the duration of response in patients with the advanced tumors. Some published studies of clinical trial reported the DBT could increase the objective response rates in a mount of tumors including advanced melanoma, non-small cell LC, RCC, and CRCA, etc. However, there are still some issues should be addressed in DBT, such as the pan-cancer predictive biomarkers have not yet been developed to identify the response state. To solve this problem, herein, we collected the first longitudinal cohort in the worldwide consisting of a total of 86 proteome profiling of plasma samples (we added more samples to 103 samples in the revision) from 22 patients with 8 different tumors.

There are some published studies focused on the biomarker exploring for immunotherapy which the cohort consisted of a variety of tumor types. For example, Pender etc. explored the biomarker of response to immune checkpoint inhibitor (ICI) in advanced solid tumors (**2021, *Clinical Cancer Research*, PMID: 33020056**) which the cohort consisted of 20 tumor types including LC, breast cancer (BRCA), stomach adenocarcinoma (STAD), CHOL, etc. Yang etc. explored the circulating tumor DNA biomarker to monitor the response of the pembrolizumab therapy across 30 cancer types (**2021, *Nature Communications*, PMID: 34446728**).

As for the response of the DBT, it is mentioned in line 132 of the previous version of manuscript “Among the 64 samples, 4 samples had partial response (iPR), 11 samples had stable disease (iSD), 4 samples had unconfirmed progressive disease (iUPD), 7 samples had confirmed

progressive disease (iCPD), and 38 samples were not-evaluated". This longitudinal DBT cohort enrolled 22 patients who received the DBT once every 3 weeks (as a therapy cycle) and were evaluated the response state to the therapy once every 6 weeks according to the iRECIST v1.1 standard. The plasma samples were collected for each patient once every 3 weeks. In order to clearly illustrate the cohort composition, as shown in **Figure RL 3A**, the scatter denoted the therapy cycle for each patient, the colored scatter (not grey) denoted the different response state, the y axis denoted the different patient (annotated with tumor type). We have updated the **Figure 1C** as **Figure RL 3A** and added the tumor type information in the legend of the **Figure 1C**.

Based on this longitudinal DBT cohort, we prefer to explore biomarkers that can predict DBT response independently of specific tumor types, namely, this biomarker could be a generalized biomarker for this strategy of DBT. Because of this DBT strategy is still in the ascendant, the incomplete tumor types and the small sample size are a limitation to further interpretation for this cohort, but we believed that these results can be served as a basis for future work and may have some clinical implications.

A

Figure RL 3A. The panel depicting the longitudinal DBT for each patient. The dot represented the evaluated state. Yellow as the pre DBT, light green as complete response (iCR), green as partial

response (iPR), blue as stable disease (iSD), pink as unconfirmed progressive disease (iUPD), red as confirmed progressive disease (iCPD), and grey as the not-evaluated.

Q6. I suggest the sample size be increased to at least >50 for both the cohort and reanalysis of the ML data.

Response:

Thanks for the insightful comments that could improve the ML model reliability. According to reviewer's suggestion, we enrolled an additional 27 samples for the discovery cohort and an additional 11 samples for the independent validation cohort. Finally, the discovery cohort contained 103 samples and the independent validation cohort contained 54 samples. We used the samples (62 samples) that have both the response state and LDH, PA, and APOC3 expression level to reconstruct the ML model. By performing the same ML pipeline as mentioned in the previous version manuscript, we reconstructed the ML model to distinguish the disease non-progressive (DNP) group and disease progressive (DP) group (**Figure RL 4A**). The ML model showed the good performance on the discovery cohort (accuracy score = 0.90). Meanwhile, this model has the good generalization ability manifested by the accuracy score was 0.93 on the independent cohort than the model of the previous version (accuracy score = 0.81).

As shown in **Figure RL 4B, C**, the confusion matrix and receiver operating characteristic curve (ROC) of train, test, and independent validation cohort showed the good performance of the classifier model. The area under ROC (AUROC) for the three cohort were 0.96, 0.96, and 0.96. Furthermore, to evaluate the model performance, we randomly split the cohort for 100 times to re-trained the model and re-evaluated the performance with more evaluation type including AUROC, accuracy, precision, recall, and F1 score. In detail, the train cohort has average scores 0.96, 0.93, 0.94, 0.93, and 0.93 with standard derivation of 0.026, 0.028, 0.029, 0.028, and 0.030, test cohort has scores 0.76, 0.81, 0.81, 0.81, and 0.79 with standard derivation of 0.13, 0.06, 0.086, 0.060, and 0.074, the independent validation cohort has scores 0.90, 0.90, 0.91, 0.90, and 0.90 with standard derivation of 0.047, 0.052, 0.031, 0.052, and 0.044 (**Figure RL 4D**). The **Figure RL 4E** showed the probability of predicted samples during the classifier decision, which indicated the good performance for distinguishing the DNP group from the DP group. We have updated these results

in **Results** part page 13 from **line 363 to 370** of the revised manuscript.

Research on DBT mechanism and biomarker exploration is in the ascendant. This study enrolled the first longitudinal plasma samples to explore biomarkers that can reflect the response state to DBT at proteomic level. Herein, we continued to collect 27 samples from patients who were enrolled in the discovery cohort and still received DBT and an additional 11 samples to the independent validation cohort, which reduced the randomness of model predictions to a certain extent and made it more robust. The reconstructed model displayed the better performance than that in the previous version for classifying the DP group from DNP group, which further indicated the features including LDH, PA, and APOC3 had the power in the therapy response prediction task.

The good performance of the ML model in this version indicated the clinical value of PA, LDH, and APOC3, which suggested enrolling more samples could help validate the findings. This cohort was the first longitudinal DBT cohort in the worldwide consisting of plasma samples designed to explore the predictive biomarker for the DBT response. Although the small sample size limited the further interpretation of the ML model, we believed these results can serve as a basis for the future work which benefit for the clinical practice. The clinical measurement and implication of these potential biomarkers offers an opportunity to expedite translation of basic research to more precise diagnosis and treatment in the clinic and improve the development of the DBT for tumors.

Figure RL 4A. Schematic of the machine learning framework including four parts as Data collection, Feature selection, Cross validation (CV), and Test cohort validation. **B.** The confusion matrix of train, test, and validation cohort of the machine learning model. **C.** The receiver operating characteristic curve (ROC) of train, test, and validation cohort of the machine learning model. **D.** The evaluated scores for 100 repeated trained model. **E.** The bar charts depicting the machine learning model's predicted probabilities for samples in the discovery and validation cohorts.

Q7. The naming of the groups is not very clear, especially the “before DBT group” – I suggest to change it to “Pre DBT group”.

Response:

We thanked for the comments and apologized for the informal naming. We appreciated the suggestion to change the name of the group “before DBT” to “Pre DBT group”. We changed the name to the “Pre DBT group” that could ensure the naming of the group is clear and concise in the revised manuscript.

Q8. The Proteomics data repository does not have any annotations on which files are pre DBT, control or 1st DBT. I request to upload the following data 1. Annotation file, 2. FASTA file used for

the analysis, 2. DIA-NN output. As a reviewer we tried to evaluate the data on and we find it stressful that the authors did not follow any proteomic community standards.

Response:

Thanks for the comments. We apologized for the deficiency of uploading data that make the reviewer stressful.

Followed by reviewer's suggestion, we have uploaded the MS related data to the ProteomeXchange Consortium via the iProX partner repository (<https://www.iprox.cn/>) under the Project ID: IPX0005695001. The link access to the data as follows:

<https://www.iprox.cn/page/PSV023.html?url=1698636112761nkY1>

The password: czy7

In detail, we uploaded the following data for the discovery cohort and independent validation cohort, separately:

- 1) Annotation file: the annotation of sample information including sample name, therapy cycle, etc.;
- 2) MS raw data: the MS raw data;
- 3) FASTA file: the FASTA file used for the MS data processing;
- 4) DIA-NN output file: the merged DIA-NN standard output file;
- 5) Proteome expression matrix: the quantification matrix data.

The more detailed description could be found in the README file under the above link. The same data uploading mode was also implemented in the previous published study (**2022, *Cell Research*, PMID: 36307579; 2023 *Cell Reports Medicine*, PMID: 37633276; 2020, *EMBO journal*, PMID: 33140861**). We believed that these data were uploaded in compliance with proteomics community standards, it can provide a convenient way to obtain data as well as ensure the quality of data interpretation and comparability of analyses. We have updated this information in **Methods** part page 24 from **line 648 to 663** of the revised manuscript.

[ ]	homo_sapiens_9606.fasta	FASTA	null	25.71M	✓
[ ]	README.txt	OTHERS	null	0.95K	✓
[ ]	Annotation file.xlsx	OTHERS	null	17.09K	✓
[ ]	DBT_DIANN_output_merge.txt.zip	SEARCH	null	13.9G	✓
[ ]	validation_DIANN_output_merge.txt.zip	SEARCH	null	5.2G	✓
[ ]	proteome expression matrix.zip	SEARCH	null	5.5M	✓
[ ]	Exp111025_HFX5_F1_R1.raw	RAW	null	643.69M	✓
[ ]	Exp174736_HFX5_F1_R1.raw	RAW	null	652.11M	✓
[ ]	Exp174740_HFX5_F1_R1.raw	RAW	null	652.32M	✓
[ ]	Exp174735_HFX5_F1_R1.raw	RAW	null	660.17M	✓
[ ]	Exp128232_HFX3_F1_R1.raw	RAW	null	713.36M	✓

Q9. Line 510-511, Quantification of peptides in DIA-NN is not the average of chromatographic fragment ion peak areas but top 6 ions.

Response:

Thanks for the comment. We apologized for the typo of the peptide quantification in DIA-NN and updated the correct description in **Methods** part as follow: “Quantification of identified peptides was calculated the sum of the intensities of the top six fragments for the reference spectra libraries.”

Q10. Calculation of Protein Quantity by Fraction of total (FOT) is not the most used way. Was the quantification done using only the unique peptides or using unique and razor peptides?

Response:

Thanks for the professional comments. In this study, we used the unique and razor peptides strategy to quantify the proteomic data. Common-used proteomic quantification software MaxQuant embedded the “unique plus razor peptides” quantification strategy (2008, *Nature Biotechnology*, PMID: 19029910). The MaxLFQ also used this “unique plus razor peptides” strategy, because it is a good compromise between the two competing interests of using only peptides that undoubtedly belong to a protein and using as many peptide signals as possible (2014, *Molecular & Cellular Proteomics*, PMID: 24942700). This strategy has been used in the published studies (2023 *Cell Reports Medicine*, PMID: 37633276; 2020, *EMBO journal*, PMID: 33140861).

Admittedly, the most used quantification strategy is the MaxLFQ strategy for the DIA data. According to reviewer's suggestion, we used the MaxLFQ strategy to quantify discovery cohort proteomic data consisting of 103 samples (updated in this version) and the independent validation cohort proteomic data consisting of 54 samples (update in this revision). Ultimately, the main conclusions were consistent with the FOT quantification strategy, such as the apolipoprotein-related biological process level and the APOC3 expression level could indicate the DBT response state.

In detail, based on the **MaxLFQ quantification**, the lipoprotein related pathway including regulation of reverse cholesterol transport ($p\text{-value} = 1.89\text{E-}3$), high-density lipoprotein particle remodeling ($p\text{-value} = 5.59\text{E-}3$), and lipoprotein lipase activity ($p\text{-value} = 1.45\text{E-}3$) showed higher level in DNP group than that in DP group (**Figure RL 5A**). Correspondingly, the apolipoprotein APOC3 ($p\text{-value} = 5.35\text{E-}5$) and APOL1 ($p\text{-value} = 9.98\text{E-}3$) showed the increased level in DBP than that in DP (**Figure RL 5C**). These results were consistent with the results based on **FOT quantification**, manifested by the lipoprotein related pathway including reverse cholesterol transport ($p\text{-value} = 1.01\text{E-}3$), high-density lipoprotein particle remodeling ($6.09\text{E-}3$) showed higher level in DNP group than that in DP group (**Figure RL 5B**); APOC3 ($p\text{-value} = 2.34\text{E-}4$) and APOL1 ($p\text{-value} = 1.69\text{E-}4$) showed the similar expression level between DNP and DP groups (**Figure RL 5D**).

In addition, we fitted the time-series linear regression model with APOC3 expression level as response variate and therapy cycle as the covariate for three patients who finally responded to the DBT (P1, P16, P18) and three patients who did not respond to the DBT (P2, P3, P22). As shown in **Figure RL 5E**, the APOC3 showed the positive correlation with the therapy cycle only for the patients who ultimately responded to the DBT (spearman rho = 0.20, 0.29, 0.50, $r^2 = 0.01, 0.39, 0.37$, $p\text{-value} = 9.23\text{E-}22, 2.16\text{E-}8, 1.76\text{E-}36$). The APOL1, APOA4, and APOC4 which were the candidate proteins in the previous version of manuscript did not display the consistent positive association with the therapy cycles. These results were consistent with the results in the previous version of manuscript.

Figure RL 5A. The dot plot shows the difference of the ssGSEA scores based on the MaxLFQ strategy between DNP and DP samples. The dot size indicates the significance derived by t-test. **B.** Same as **Figure RL 5A** but based on the FOT strategy. **C.** The distribution of lipoprotein APOC3 and APOL1 for DNP and DP groups based on the MaxLFQ strategy. **D.** Same as **Figure RL 5C** but based on the FOT strategy. **E.** The panel shows the time series-linear regression of APOC3, APOC4, APOA4, and APOL1 in 6 patients (P1, P18, P16 with all DNP samples, and P22, P3, P2 with DP samples). Dot color represents the evaluated response state. The top panel was the result based on the MaxLFQ strategy and the bottom was the result based on the FOT strategy.

The key point of the cohort research is to find out the proper biomarker that could indicate the response state during the DBT cycle. We also reconstructed the machine learning model based on the MaxLFQ quantification strategy. As shown in **Figure RL 5F**, the receiver operating

characteristic curve (ROC) and the confusion matrix of the test, and independent validation cohort showed the good performance of the classifier model. The area under ROC (AUROC) for the two cohort were 0.99 and 0.97. Furthermore, to evaluate the model performance, we randomly split the cohort for 100 times to re-trained the model and re-evaluated the performance with more evaluation type including accuracy, precision, recall, and F1 score. In detail, the train cohort has average scores 0.82, 0.85, 0.85, and 0.85 with standard derivation of 0.048, 0.037, 0.033, 0.037, and 0.034, test cohort has scores 0.75, 0.81, 0.81, and 0.80 with standard derivation of 0.14, 0.075, 0.083, 0.075, and 0.077, the independent validation cohort has scores 0.97, 0.92, 0.93, 0.92, and 0.92 with standard derivation of 0.0063, 0.013, 0.014, 0.014, and 0.013 (**Figure RL 5G**). The **Figure RL 5H** showed the probability of predicted samples during the classifier decision, namely the specificity was 0.93 for the discovery cohort and 0.96 for the validation cohort, which indicated the good performance for distinguishing the DNP group from the DP group. These results showed the ML model based on MaxLFQ quantification strategy has the same good performance compared with the ML model based on FOT quantification strategy (details about the ML results could be found in the R2-Q6).

We have updated these results in **Figure S9**, in **Discussion** part page 17 **from line 470 to 498** of the revised manuscript.

Figure RL 5F. The receiver operating characteristic curve (ROC) and the confusion matrix of test and validation cohort of the machine learning model. **G.** The evaluated scores for 100 repeated trained model. **H.** The bar charts depicting the machine learning model's predicted probabilities for samples in the discovery and validation cohorts.

Q11. Calculating MaxLFQ may help reduce the CVs even further.

Response:

Thanks for the insightful comments. According to reviewer's suggestion, we used MaxLFQ as the proteome quantification. The detailed information of MaxLFQ could be found in the response of the R2-Q10.

Moreover, we calculated the MaxLFQ for the quality control (QC) samples and evaluated its coefficient of variance (CV). In detail, as shown in **Figure RL 6A**, the CVs for the QC sample under the MaxLFQ algorithm (range from 0.1% to 25.3%) significantly lower than that under the raw

intensity quantification strategy which used in the previous version of manuscript (range from 2.84% to 214.47%). Furthermore, we compared the CVs between the MaxLFQ strategy and the raw intensity quantification strategy. The CVs under two strategies showed the similar distribution (right-skewed) (drew in 10 bins), but the MaxLFQ strategy showed the smaller skewness than that in raw intensity quantification strategy (1.75 less than 3.49). It is worth noting that under the MaxLFQ strategy, the proteins with less CVs have the larger proportion than that under the raw intensity quantification strategy, which indicated the MaxLFQ algorithm could help reduced the CV as reviewer mentioned (**Figure RL 6A**).

Figure RL 6A. The bar plot shows the distribution of the CVs between MaxLFQ strategy and previous strategy. The left panel was the result based on the MaxLFQ strategy and the right was the result based on the raw intensity quantification strategy.

Q12. The result file in the repository has only one column with FOT, were all time points merged?
This kind of reporting is seen in a black box.

Response:

Thanks for the comments. We apologized for the deficiency of uploading data that make reviewer confused and stressful. The result file in the repository was not the FOT value that merged all time points. In order to provide a convenient way to obtain data, we re-uploaded the proteome quantification matrix of the discovery cohort to the iProX. The detailed information could be found in the response of R2-Q8.

[ ]	homo_sapiens_9606.fasta	FASTA	null	25.71M	✓
[ ]	README.txt	OTHERS	null	0.95K	✓
[ ]	Annotation file.xlsx	OTHERS	null	17.09K	✓

[ ]	DBT_DIANN_output_merge.txt.zip	SEARCH	null	13.9G	✓
[ ]	validation_DIANN_output_merge.txt.zip	SEARCH	null	5.2G	✓
[ ]	proteome expression matrix.zip	SEARCH	null	5.5M	✓

Q13. It is not very clear what protein abundance value the authors use. Is it fraction of total value calculate for each sample? Or is it iBAQ value?

Response:

Thanks for the comments. We apologized for the misleading description that makes reviewer confused.

Herein, the proteome quantification was conducted using the intensity-based absolute quantification (iBAQ) algorithm (2011, *Nature*, PMID: 21593866), followed by the fraction of total (FOT) normalization as reported previously (2023, *Cell Reports Medicine*, PMID: 37633276; 2018, *Nature Communications*, PMID: 29739932). In detail, the FOT was the ratio of iBAQ for the specific protein against to all proteins in one sample, in order to avoid the effects of the extreme minimum values in the downstream analysis, we multiply the 1E10 to each of ratio value as the ultimate FOT value. The published studies that enrolled plasma samples also adopted the same quantification strategy (iBAQ combined with FOT) (2020, *EMBO journal*, PMID: 33140861; 2022, *Cell Research*, PMID: 36307579).

For the DIA data quantification, the most used quantification strategy is the MaxLFQ strategy, in order to improve the data reproducibility and validate the results that found in this cohort, we used the MaxLFQ strategy to quantify discovery cohort proteomic data in the revision. The detailed information could be found in the response of R2-Q10.

We have updated this information in **Methods** part page 20 from **line 564 to 568** of the revised manuscript. Thanks again for the point out.

Q14. Lines 220-221, Grouping of iPR and iSD could be changed to “disease non-progressive” and not, “non-disease progression”.

Response:

Thanks for the comment. We have changed the name of the grouping of iPR and iSD to “disease non-progressive (DNP)” instead of “non-disease progression (DNP)” in the revised manuscript.

Q15. Line 197, it is good to indicate fold changes instead of p values for particular proteins.

Response:

Thanks for the comments. According to reviewer’s suggestion, for line 207, we have added the fold changes apart from the p values as “The increased level of ACAT, the key enzymes responsible for the formation of cholesteryl esters, in HLH protein cluster might imply the increased level of cholesterol in DBT patients’ plasma (ACAT1 p-value = 7.07E-3, fold change = 4.95; ACAT2 p-value = 0.03, fold change = 2.41, **Figure 2C**).”

In addition, we have added the fold change information to the proper circumstance to make the result more definite and unambiguous. For example, we changed line 260 to “Besides, at proteome level, we found the protein CTSL, participating in the release of thyroid hormone, was upregulated in DNP (**Figure 3G**, t-test p-value = 0.031, fold change = 1.36)” and changed line 283 to “In detail, the tubulin-related processes in DNP group were manifested by the upregulation of TUBB4B, TUBB8, TUBB2B, and TUBB2A (t-test p-value = 7.90E-4, 2.50E-3, 4.27E-3, 4.27E-3, fold change = 2.14, 2.51, 2.09 and 2.06, **Figure S5A, Supplementary Table 4**)”.

Q16. Line 177-179, The authors used $p\text{-value} \leq 0.05$ as a criterion for significance, it is highly recommended to apply an additional FDR control.

Response:

Thanks for the professional comments. According to reviewer’s suggestion, we used the additional control as $FDR < 0.05$ and re-analyzed the plasma proteome characteristics among healthy control group, Pre DBT group and 1st DBT cycle group. The results were consistent with the $p\text{-value} < 0.05$ criterion.

As shown in **Figure RL 7A** (compared with the **Figure RL 7C** under the p -value < 0.05 criterion), the results revealed alterations in the proteome composition, manifested as 240 significantly differently expressed proteins (DEPs). We then performed the same analysis pipeline to define the four protein clusters and used the over-represented analysis (ORA) for each of protein cluster. In detail, we classified the proteins as high (H), medium (M) and low (L), and grouped DEPs into four clusters (annotated as HLH, LMH, LHL, HML) by comparing the relative z-score value among three sample groups (healthy control, Pre DBT group and 1st DBT cycle group).

As shown in **Figure RL 7B** (compared with the **Figure RL 7D** under the p -value < 0.05 criterion), LHL protein cluster was characterized by extracellular matrix organization, L1CAM interactions, and PI3K signaling events mediated by AKT (YWHAB, CHL1, ITGAV), indicating the “calm down” of tumor-associated proteins after the DBT. Proteins in HLH cluster, implying the recovery of tumor-inhibited protein expression after the DBT, participated in cofactor catabolic process and NADH regeneration (ACAT1, HK1, ADPGK). LMH cluster showed significantly enrichment of immune-related processes such as innate immune system and complement cascade (SAA1, SAA2, FGA). The **Figure RL 7D** showed the consistent results of the enriched biological processes.

Compared to the ORA result of the previous version of manuscript, the statistical rigorous criterion gives the similar results, which indicated the enriched biological processes such as the PI3K signaling event, extracellular matrix and immune system processes are the robust result during the DBT therapy. We have updated these results in **Results** part page 7 from **line 187 to 203** of the revised manuscript.

Figure RL 7A. Comparison of plasma protein changes before and after DBT treatment. The x-axis indicates the comparison of pre DBT group and 1st DBT cycle group, and y-axis indicates the comparison of healthy control group and pre DBT group. The color represents the significance of protein (FDR for p -values derived by Kruskal–Wallis’s test). **B.** The bubble plot depicting the pathway enrichment of three protein groups (HLH, LHL, LMH). Dot size represents the protein counts of the pathway and color represents the pathway enrichment significance. **C.** Same as **Figure RL 7A**, but under the p -value < 0.05 criterion. **D.** Same as **Figure RL 7B**, but under the p -value < 0.05 criterion.

Q17. Line 181, I suggest to provide the z-score values as an additional supplementary table.

Response:

Thanks for the comments. According to reviewer’s suggestion, we have provided the z-score value for each of healthy control, Pre DBT group and 1st DBT cycle group in the **Supplementary Table 2a** in the revision.

Q18. Figure 2 A, X and Y axis titles should be included in the figure instead of explaining in the legend separately. As it makes it difficult to understand if the ratios are before DBT/1st DBT or 1st DBT/before DBT.

Response:

Thanks for the professional comments. We apologized for the confused annotation in the figure.

According to reviewer’s suggestion, we added the interpretation of the x and y axis as “1st DBT group vs. Pre DBT group” and “healthy control vs. Pre DBT group” to make the figure more accessible (**Figure RL 8A**). We have updated these results in **Figure 2A** and the corresponding **Legend** in the revised manuscript.

Figure RL 8A. Comparison of plasma protein changes before and after DBT treatment. The x-axis indicates the comparison of pre DBT group and 1st DBT cycle group, and y-axis indicates the comparison of healthy control group and pre DBT group. The color represents the significance of protein (FDR for *p-values* derived by Kruskal-Wallis’s test).

Q19. In Supplementary figure 1, the authors have shown the correlation and CV values for the QC samples only, they could also provide the same for the actual samples.

Response:

Thanks for the professional comments. In order to evaluate the reproducibility of the MS detection for the actual samples, we had randomly selected 10 actual samples and detected in consecutive replicates 5 times for each sample. We apologized for not providing this part in the previous version of the manuscript. The detailed analysis information was as follows.

We randomly selected 10 actual samples and performed the same sample preparation pipeline. Each sample was detected in consecutive replicates 5 times using the same MS method as for the actual samples of the DBT cohort. We calculated the coefficient of variance (CV) for each of the sample among the 5 repeats. As shown in **Figure RL 9A, B**, the distribution of the CVs showed the

similar distribution and the median of CVs for each of sample ranged from 11% to 15% (range from 1.0% to 1.3% under the MaxLFQ strategy). **Figure RL 9C** showed the CVs of proteins in each of samples (blue dot indicated the CVs less than 30%). In addition, we calculated the pearson correlation among the 5 repeats for each of samples. As shown in **Figure RL 9D**, the correlation ranged from 0.94 to 0.98, which indicated the similar quality control level with the previously published studies (**2019, *Nature*, PMID: 30814741; 2016, *Cell Systems*, PMID: 27135364**).

Collectively, the high correlation and low CVs among the repeated actual samples indicated the stability and robustness of the MS platform. Based on these results, therefore, we detected once for each sample in the actual DBT cohort as the previous study reported strategy (**2019, *Nature*, PMID: 30814741**). We have updated these results in **Figure S1, Results** part (page 6, **line 152 to 159**) and **Methods** part (page 21, **line 571 to 575**) of the revised manuscript.

Figure RL 9A and B. The density plot of protein CVs for each of 10 samples. **C.** The scatter plot depicting the CVs among the 5 repeats for each of 10 samples. The blue dot indicated the CVs less than 30%. **D.** The heatmap of the Pearson correlation among 5 repeats for each of 10 samples.

Q20. Line 356, No proteomics data for the validation cohort has been uploaded to the repository. I recommend to upload this data as well with the additional data mentioned in point 3.

Response:

Thanks for the professional comments. According to reviewer's suggestion, we have uploaded

the proteomics data including MS raw data, DIA-NN output, and the proteome expression matrix for the discovery cohort and validation cohort to the iProX repository. More details can be found in the response of the R2-Q8.

[ ]	homo_sapiens_9606.fasta	FASTA	null	25.71M	✓
[ ]	README.txt	OTHERS	null	0.95K	✓
[ ]	Annotation file.xlsx	OTHERS	null	17.09K	✓
[ ]	DBT_DIANN_output_merge.txt.zip	SEARCH	null	13.9G	✓
[ ]	validation_DIANN_output_merge.txt.zip	SEARCH	null	5.2G	✓
[ ]	proteome expression matrix.zip	SEARCH	null	5.5M	✓
[ ]	Exp111025_HFX5_F1_R1.raw	RAW	null	643.69M	✓
[ ]	Exp174736_HFX5_F1_R1.raw	RAW	null	652.11M	✓
[ ]	Exp174740_HFX5_F1_R1.raw	RAW	null	652.32M	✓
[ ]	Exp174735_HFX5_F1_R1.raw	RAW	null	660.17M	✓
[ ]	Exp128232_HFX3_F1_R1.raw	RAW	null	713.36M	✓

Q21. Supplementary 5 and 6 does not serve the purpose for this study. What is the point of having this scan while the study is not to assess the DBT response outcome.

Response:

Thanks for the comments. We apologized for the unclear description of the importance about the radiology imaging in the biomarker discovering. We responded this comment into two parts including (1) the clinical response evaluation and (2) the integration of imaging and proteomic data.

As for the clinical response evaluation, the radiology imaging is the key strategy to evaluate the tumor size and thereby obtain the therapy response state (2021, *Journal of Nuclear Medicine*, PMID: 33334912; 2019, *Radiology*, PMID: 29782246). Herein, the response for the DBT for each patient was determined by the radiology imaging and other strategy after receiving two therapy cycles. The radiology imaging could intuitively reflect the response state.

As for the integration of imaging and proteomic data, herein, in order to directly reflect the association between clinical radiology imaging and the potential biomarker derived from the proteomic data, we evaluated the correlation between APOC3 expression level and the radiology

imaging at qualitative (**Figure S5** and **S6** in the previous version of manuscript) and quantitative (**Figure 4F** in the previous version of manuscript) level, which reflect the importance of APOC3 to indicate the response state during the DBT. These results directly reflect the clinical value of APOC3 in the clinical trial. The same strategy was also utilized in the published research, for example, Jiang et. found a subtype (BLIS with low Homologous Recombination Deficiency score) of triple negative breast cancer (TNBC) defined by the transcriptome data with worse prognosis and lacks immune activation, in addition, they reported the radiology imaging of the patients in this subtype who received the chemotherapy or radiotherapy to prove the worse prognosis conclusion (**2019, *Cancer Cell*, PMID: 30853353**).

In summary, the radiology imaging is an additional independent aspect to help us to examine what has been discovered based on the plasma proteome data.

We have updated this information in **Discussion** part page 17 from **line 470 to 498** of the revised manuscript.

Reviewer #3 (Remarks to the Author): expert in proteomics model development

The team sought to examine the biomarkers for dual blocker therapy in a proteomics study. The team is known for the proteomics platform and has built track records on proteomics studies.

Q22. Although the study reports interesting findings on proteomic profiles after DBT treatment, the clinical relevance of the findings should be strengthened. If the biomarkers for DBT response can be used as "intermediate" biomarkers for the survival of patients or other long-term outcome after DBT response, the clinical significance may be better justified.

Response:

Thanks for your insightful comments.

Herein, to explore the biomarker that could indicate the response state during the dual blocker therapy (DBT), we collected the plasma samples and performed the proteomic analysis. We observed the cholesterol related biological processes showing increased level in the non-disease progression (NDP) group (Since the informal nomenclature was changed to the disease **non-**

progressive (DNP) group in this revision according to the suggestion of Reviewer 2) than that in disease progressive (DP) group. In addition, by performing the line regression, we found the APOC3 showed the best performance than other cholesterol related proteins in predicting the response state. Combined with the clinical indicator prealbumin (PA) and lactate dehydrogenase (LDH), we constructed the machine learning model with good performance and validated on the independent cohort (due to add more samples for both DBT cohort and independent validation cohort, the updated results of ML model were in **Figure 5** in the revised manuscript).

According to reviewer's suggestion, we constructed the ML model with features as the "intermediate" biomarker LDH, PA, and APOC3 to predict whether the patient could sustained respond to DBT or not.

Considering the avoidance of the overfitting risk for the machine learning model, we came up with three principals as (1) Using as many samples as possible to train the model; (2) Using as simply hyperparameters as possible to train model; (3) Using the ensemble strategy to improve the prediction confidence. Based on the three principals, we designed the pipeline as four parts: sample selection, feature selection, model training and testing, and model performance evaluation.

As for the sample selection part, we preferred this machine learning model can provide the accurate predictions at early stages of treatment. Hence, we selected the 1st DBT cycle samples and 2nd DBT cycle samples, separately. **As for the feature selection part**, we used the clinical indicator LDH, PA and proteomic feature APOC3. **As for the model training and testing part**, we used the leave one out strategy (python scikit-learn.model_selection.LeaveOneOut) (version 1.1.3) to split the train and test cohort, and used all of the combination to train modeling according to the principal (1). We used the 21 state-of-the-art machine learning models (AdaBoost, Bagging, Extra Trees, Gradient Boosting, Random Forest, Gaussian Process, Logistic regression, Passive Aggressive, Ridge classifier, SGD classifier, Perceptron, Bernoulli naïve bayes, Gaussian naïve bayes, KNeighbors, SVC, NuSVC, Decision Tree, Extra Tree, Linear Discriminant, Quadratic Discriminant, and XGBoost) to train the ensemble model according to the principal (3). **As for the model performance evaluation part**, because of using the ensemble strategy, we proposed a

weighted prediction strategy for the 21 model predictions. In detail, we calculated decision scores followed by the formular:

$$decision\ score_j = sign\left(\sum_{i=1}^{21} predict\ label_{ij} \times \frac{balanced\ accuracy\ score_{ij} - 0.5}{\sum_{i=1}^{21} balanced\ accuracy\ score_{ij} - 0.5}\right)$$

For each j sample by i machine learning model

The sample was predicted as DNP if the decision score = 1, otherwise predicted as DP. As shown in **Figure RL 10A**, this ensemble model showed good performance based on the 1st DBT cycle (balanced accuracy score = 0.84), and 2nd DBT cycle samples (balanced accuracy score = 0.72). Notably, the 1st DBT cycle has greater potential to determine whether the patient could sustained respond to DBT or not.

In summary, the clinical and proteomic features which explored by the longitudinal DBT cohort also have the ability to predict whether the patient could sustained respond to DBT or not at the early stage of the DBT based on a simply ML model pipeline. This result indicated the potential prognosis clinical values of these three features (LDH, PA, and APOC3) that could predict the long-term response to the DBT for patients. We have added these results in the **Results** part from page 14 **line 383 to 398** and **from 405 to 407**; and **Methods** part page 23 from **line 619 to 642** of the revised manuscript.

Figure RL 10A. The heatmap depicting the true label and the predicted label of the cohort. The left panel was the result of the ML model that trained and tested under the 1st DBT cycle samples, the

right panel was the result under the 2nd DBT cycle samples.

Q23. Following the last comment, if the authors can better justify the clinical application of their findings, how frequently would physicians perform proteomics study to monitor the response? How would physicians read the interpret the findings?

Response:

Thanks for the professional comments.

Based on the results described in the response of R3-Q22, we found the 1st DBT samples have the better performance for the prediction task. Therefore, we recommended physicians pay more attention on the patients who have received the 1st DBT to predict whether the patient could sustained respond to DBT or not. Furthermore, we noticed the patient 21 and patient 23 were wrongly classified as DNP and DP at the 1st DBT cycle, and patient 21, patient 3 and patient 14 were wrongly classified as DNP, DP, and DP. This result reminded us for the prediction task of whether the patient could sustained respond to DBT, the prediction of the 2nd DBT cycle could serve as the complementary for the 1st DBT cycle. We have added these results in the **Results** part from **line 383 to 398** of the revised manuscript.

Based on the simply ensemble ML model, physicians could directly obtain the state of whether the patient could sustained respond to DBT or not. This ML pipeline provided a robust method to predict the final response state and, in the future, we will collect a large cohort to validate the performance of this strategy, which can more directly guide the clinical practice.

REVIEWER COMMENTS

Reviewer #1 (Remarks to the Author):

Thanks to the authors for addressing all comments in great detail.

Reviewer #2 (Remarks to the Author):

In the revised version of the manuscript by Lyu and colleagues, efforts were made to address my comments; however, they fell short of achieving a satisfactory level of resolution. Notably, concerns persist regarding data analysis and methodological details, which warrant further attention and clarification.

One key concern pertains to the variation in protein concentration within plasma (ranging from 30µg to 60µg per µl). The method section lacks explicit information on the quantity of samples injected per analysis, a crucial detail given the authors' focus on the differential expression of APOC3 as a marker. To enhance the reliability of their findings, I recommend the utilization of MaxLFQ over iBAQ or FOT, especially considering the importance of accurately assessing protein expression.

Moreover, while the authors mentioned employing a total of 327 libraries as reference spectra libraries, the method section lacks sufficient details on the generation and utilization of this library. Does the 327 libraries derived from plasma proteome? The overall conclusion drawn from the differential expression of APOC3 as a marker is based on a relatively small sample size of 22 and 27 patients, raising concerns about the robustness of the conclusions.

The description of the Maxquant and DIANN tools is incomplete, as the full reports of these searches were not provided. I obtained their submitted data from the specified link (<https://www.iprox.cn/page/PSV023.html?url=1698636112761nkY1>). The authors must present a comprehensive overview of their analysis tools for transparency and reproducibility.

The authors' response in the method section, particularly in quantifying identified peptides, appears insufficient. The use of MaxLFQ values, which demonstrate better Coefficient of Variation (CoV), is not adequately justified compared to FOT or iBAQ. The clarification provided regarding the quantification process using iBAQ, FOT, and the associated normalization procedures lacks clarity and needs further elucidation.

Furthermore, the data preprocessing methods deviate from proteomic community standards, as the manuscript currently allows for 30% valid values in total for both cohorts. I recommend revising this to adhere to the community standard of 70% valid values for enhanced reliability and process the downstream analysis.

The imputation strategy applied must also be thoroughly explained, and specific values should be provided.

Concerns persist regarding the DIANN tool, which lacks a default option for iBAQ. The method section fails to specify how iBAQ values were extracted from the DIA-NN tool, necessitating clarification for transparency and reproducibility.

FDR permutation based two sample t-test (DNP vs DP, FDR 0.05) finds only 14 proteins significantly regulated in discover cohort (ACAN, HCY, APOC2, APOC3, AZGP1, CD44, CNDP1, F7, GFAP, HSPD1, IGHV1-69, IGHV2-70D, ITIH2, PAICS, RBP4) and no significance in validation cohort.

The discrepancy between the number of significantly regulated proteins in the discovery cohort and the lack of significance in the validation cohort raises doubts about the reproducibility of the findings. Figures 3G, 4C, Supplementary Figure 5A, and Supplementary Figure 9B are flagged as not significant and, therefore, not reproducible.

Reviewer #3 (Remarks to the Author):

The authors have responded to the comments satisfactorily.

RESPONSE TO REVIEWERS' COMMENTS

In the revised version of the manuscript by Lyu and colleagues, efforts were made to address my comments; however, they fell short of achieving a satisfactory level of resolution. Notably, concerns persist regarding data analysis and methodological details, which warrant further attention and clarification.

Response:

Thank you very much for the comments about the data analysis and methodological details. We thank the reviewer for the constructive and insightful comments, which help to improve the quality of this manuscript. Herein, we summarized the reviewer's comments as following: (1) about the description for the quantity of samples injected per analysis during the MS detection; (2) about the DIA data including the library construction and the MaxLFQ quantification strategy; (3) about the details for the data preprocessing including the valid value cutoff and imputation; (4) about the analysis result reproducibility and reliability. According to reviewer's comments, we made corresponding revision to address all the points as follows.

Q1. One key concern pertains to the variation in protein concentration within plasma (ranging from 30 μ g to 60 μ g per μ l). The method section lacks explicit information on the quantity of samples injected per analysis.

Response:

Thanks for the comments. We apologize for the unclear description of the protein concentration and information on the quantity of samples injected per analysis in the **Methods** parts. Proteins circulating in the human bloodstream can provide a glimpse into an individual's state of health. The dynamic nature of the plasma proteome and the accessibility of human blood makes these proteins valuable tools for diagnosing and predicting disease, identifying therapeutic targets and elucidating disease pathophysiology (2021, *Nature Reviews Genetics*, PMID: 32860016).

In order to investigate whether there were systemic differences in gross protein

concentration between different groups in dual blocker therapy (DBT) cohort (such as the disease non-progressive (DNP) and disease progressive (DP) group), we analyzed the recorded protein concentrations measured during sample preparation. In this study, plasma was separated by centrifugation at 16,000 g at 4 °C for 10 min to remove insoluble solids for proteomic analysis. Firstly, the protein concentration was measured by pipetting 2 µL of plasma sample, adding it to 98 µL of 50mM NH₄HCO₃. The diluted solution was thoroughly mixed by pipetting 50 times up and down, for a volume of 100 µL. The protein concentration of each sample was measured and recorded using the BCA method on a NanoDrop (Thermo Fisher Scientific) at 562 nm absorbance. Secondly, after sample preparation, the total amount of peptides was measured by redissolving the peptide sample in 50 µL buffer A (0.1% formic acid). The peptide concentration of each sample was measured and recorded using a NanoDrop (Thermo Fisher Scientific) at 280 nm absorbance, and ultimately a standard loading amount of 200 ng peptide for the proteomic measurement.

We performed statistical analysis on the plasma protein concentration of samples in this DBT cohort (2017, *Nature Protocol*, PMID: 28749931; 2020, *EMBO Journal*, PMID: 33140861). We observed that the average protein concentration of DNP samples is 69.05 µg/µL, and the average protein concentration of DP samples is 69.41 µg/µL, which is close to the commonly used clinical reference range for plasma protein (approximately 60-80 µg/µL) (2023, *Biomolecules*, PMID: 36979342). Further statistical analysis revealed the protein concentration between the DBP and DP samples had no significant difference (*p-value* = 0.53, fold change (DP/DNP) = 0.99).

We have updated the protein concentration table, the details of the sample preparation, and the quantity of samples injected per analysis descriptions to **Supplementary Table 1** and **Methods** part (from line 478 to line 480 and line 500 to line 504) in the revised manuscript, respectively.

Figure RL 1A. The distribution of the protein concentration between DNP and DP group. *P-values* were derived by the permutation-based two sample t-test.

Q2. A crucial detail given the authors' focus on the differential expression of APOC3 as a marker. To enhance the reliability of their findings, I recommend the utilization of MaxLFQ over iBAQ or FOT, especially considering the importance of accurately assessing protein expression.

Response:

Thanks for the professional comments about the quantification strategy utilization for the proteome data, which helps ensure the accuracy of proteomic data quantification of this DBT cohort.

The intensity based absolute quantification (iBAQ) value is a common-used quantification strategy in proteome, especially in the data-dependent analysis (DDA) proteome (2011, *Nature*, PMID: 21593866; 2015; *Nature Neuroscience*, PMID: 26523646; 2019, *Nature*, PMID: 30814741). iBAQ determines as the sum of all peptide peak intensities divided by the number of theoretically observable tryptic peptides. Hence, the iBAQ value could make the different proteins in one sample more comparable. Admittedly, the most used quantification strategy is the MaxLFQ for the DIA data. According to reviewer's suggestion, in this revised manuscript, we utilized

the MaxLFQ strategy to quantify discovery cohort proteomic data consisting of 137 samples and the independent validation cohort proteomic data consisting of 54 samples. The MaxLFQ was calculated by the 'iq' R library (<https://github.com/typham/iq>) (2020, *Bioinformatics*, PMID: 31909781). Furthermore, according to the Q7 and Q10, the proteins identified in less than 70% of samples were excluded and the permutation-based t-test statistic strategy was used for the downstream analysis. We re-analyzed the proteome data and found the main conclusions were remain unchanged, such as the apolipoprotein-related biological process level and the APOC3 expression level could indicate the DBT response state (Figure RL 4).

In order to illustrate the results and conclusions that based on the MaxLFQ quantification more clearly, we described the main findings from four aspects, including (1) the effects of DBT on plasma proteome of patients (corresponded to Figure 2 in the revised manuscript); (2) the clinical indicators alteration between disease progressive (DP) group and the disease non-progressive (DNP) group (corresponded to Figure 3 in the revised manuscript); (3) the proteome characteristics alteration between DNP and DP (corresponded to Figure 4 in the revised manuscript); (4) the machine learning predictive model for the classification of DBT (corresponded to Figure 5 in the revised manuscript).

As for the effects of DBT on plasma proteome of patients

Our initial aim was to investigate the effects of DBT on plasma proteome of patients. Therefore, we compared plasma proteome characteristics among healthy control group, pre DBT group, and 1st DBT cycle group. As shown in Figure RL 2A, the results revealed dramatic alterations in the proteome composition, manifested as 82 significantly differently expressed proteins (DEPs) (Kruskal-Wallis test FDR < 0.05). We defined the protein expression state as high (H), medium (M), and low (L), and then grouped DEPs into four protein clusters (annotated as HLH, LMH, LHL, and HML) by

comparing the relative z-score value among three sample groups (healthy control, pre DBT, and 1st DBT cycle group). For example, in the LMH protein cluster, the healthy control group had low protein levels (L), the pre DBT group had moderate protein levels (M), and the 1st DBT cycle group had high protein levels (H).

Over-represented analysis (ORA) was performed to explore the enriched biological processes for the four protein clusters. Proteins in HLH cluster, implying the recovery of tumor-inhibited protein expression after the DBT, participated in lipid binding and antioxidant activity (APOM, CETP, and ALB). LHL protein cluster was characterized by extracellular space (HRG, HGFAC), indicating the “calm down” of tumor-associated proteins after the DBT. LMH cluster showed significantly enrichment of immune-related processes, such as lymphocyte mediated immunity and complement cascade (SAA1, FGA, and CD14) (**Figure RL 2B, C**). These results represented the strong impacts of DBT on the cholesterol processes, immune response, and oncogenic signaling. These findings including the lipid, extracellular, and immune-related processes were consistent with that described in the previous manuscript under the iBAQ quantification strategy.

ICIs can lead to a distinct constellation of organ-specific side effects. In this study, we wondered whether the DEPs showed the enrichment of organ specific protein expression pattern, which indicated whether the DBT primarily affected the specific organ. Hence, we compared the above four protein clusters with the HPA dataset. As a result, there were 36 DEPs annotated as liver-specific protein (Chi square test *p-value* < 0.05, **Figure RL 2D and 2E**), followed by 4 DEPs were annotated as blood and immune system-specific. Meanwhile, we evaluated the level of liver-function indicators between pre DBT group and 1st DBT cycle group samples. The levels of aspartate transaminase (AST), Alanine aminotransferase (ALT), alkaline phosphatase (ALP), and Gamma-glutamyl transferase (GGT), which reflected the liver function damage, were

elevated in pairwise samples with 11%, 24%, 13%, and 50% increasement (**Figure RL 2F**). These results indicated that the liver function damage should be considered in DBT clinical trials, and the linkage between DBT and liver function should be further investigated. These results were consistent with that under the iBAQ quantification strategy in the previous version of manuscript. We have updated these related results from line 184 to line 186, line 193 to line 202, and line 206 to line 209 in the revised manuscript.

Figure RL 2 A. Comparison of plasma protein changes before and after DBT treatment. The x-axis indicates the comparison of pre DBT group and first DBT group, and y-axis indicates the comparison of healthy control group and pre DBT group. The color represents the significance of protein (Kruskal-Wallis test FDR). **B.** The bubble plot depicting the pathway enrichment of three protein groups (HLH, LHL, LMH) which defined in **RL 2A**. Dot size represents the protein counts of the pathway and color represents the pathway enrichment significance. **C.** Sample-wise plot of proteins involved in the pathway which described in **RL 2B** between pre DBT samples and first DBT samples. *P-value* was derived by paired t-test. **D.** The stacked bar plot depicting the overlap of differently expressed proteins and the tissue-specific proteins from HPA

dataset. *P-value* was calculated by chi square test. **E.** The heatmap of 27 differently expressed proteins (HLH, LHL, LMH) which annotated as liver-specific proteins among health control group, pre DBT group, and 1st DBT cycle group. Values were transformed by z-score. **F.** Pair-wised level of four liver function damage related blood routines between pre DBT and first DBT samples. AST, Aspartate transaminase; ALT, Alanine aminotransferase; ALP, alkaline phosphatase; GGT, Gamma-glutamyl transferase.

As for the clinical indicator alteration between DP group and DNP group

As described in the previous version of manuscript, in order to deeply illustrate the clinical indicators and proteome characteristics alteration of DBT, we classified iUPD and iCPD samples as DP group, iPR and iSD as DNP group, and compared the clinical indicators between the two groups. As a result, lactate dehydrogenase (LDH), alkaline phosphatase (ALP), and white blood count (WBC) were significantly upregulated in DP group and hemoglobin (HGB), red blood cell (RBC), free triiodothyronine (FT3), albumin globulin ratio (A/G), prealbumin (PA, also known as thyroxine-binding prealbumin) level, and Creatine Kinase-MB (CK-MB) were upregulated in DNP group (permutation-based t-test FDR < 0.05) (**Figure RL 3A**). Next, we found it only for PA and LDH that the majority of DNP sample levels were within the reference interval, while a part of DP sample levels was not (**Figure RL 3B**). In addition, we tested the significantly different clinical indicators including LDH and PA at pair-wised level by comparing the pre DBT group and the DP group. As a result, only PA was significant different in DP samples than matched pre DBT group samples (pair-wise t-test *p-value* = 0.024, **Figure RL 3C**). In this revision, although we chose the strict statistical method as the permutation-based two sample t-test, the differences of clinical indicators between DP and DNP were consistent with that in the previous manuscript. This result indicated the PA and LDH were robust for indicating the response state to DBT.

PA, also known as thyroxine-binding prealbumin or transthyretin, is a transport protein in the plasma and cerebrospinal fluid that transports the thyroid hormone thyroxine and retinol to the liver. In our cohort, the thyroid hormone FT3 and triiodothyronine (T3) showed higher level in DNP than DP samples (**Figure RL 3D**, permutation-based t-test p -value = 1.00E-4 and 5.40E-3). The function of FT3 is similar to thyroxine, but with faster and stronger physiological effects. Furthermore, we found the extremely significantly positive correlation between FT3 and PA (**Figure RL 3E**, pearson corr = 0.50, p -value = 1.24E-3), which indicated attenuation of thyroid hormone synthesis in DP samples. It is well-established that thyroid hormone participates in lipid and glucose metabolism, and the neuron development (**2014, *Physiological Reviews*, PMID: 24692351**). To figure out the downstream biological processes of thyroid hormone in DBT response, we calculated the lipid-related, glucose-related, and neuron-related scores by ssGSEA and performed spearman correlation analysis with FT3. By comparing the correlation coefficients between lipid, glucose, and neuron related processes, we found lipid-related score was most correlated with FT3 (permutation-based t-test, lipid-related vs. glucose-related p -value = 2.00E-4 and neuron-related permutation-based t-test p -value = 0.032) (**Figure RL 3F**). In addition, we classified the lipid-related processes as phospholipid, glycerophospholipid, sphingolipid, cholesterol, fatty acid, and ketone bodies. The result showed the cholesterol has the highest correlation with the FT3 (**Figure RL 3G**). This result implied the important role of cholesterol metabolism in DBT response. In this revised manuscript, under the MaxLFQ quantification strategy, we still observed the high correlation between thyroid hormone and lipid-related, especially the cholesterol-related biological processes. This result indicated the cholesterol-related biological processes were the valuable and robust for indicating the response state to DBT. We have updated these related results from line 240 to line 246, line 262 to line 271, and line 251 to line 257 in the revised manuscript.

Figure RL 3 A. Heatmap depicting the significant different level of blood routines. FDR were derived from the adjusted permutation-based t-test p-values. Values were transformed by z-score. **B.** The boxplot describes the level of prealbumin (PA) and lactate dehydrogenase (LDH) between disease non-progressive (DNP) and disease progressive (DP) samples at blood routine level. The pink rectangle shows the normal reference interval. The line represents median, upper, and lower quartiles, respectively. *P-value* was derived by permutation-based t-test. **C.** Pair-wised plot of PA and LDH at blood routine level between pre DBT samples and DP samples. *P-value* was derived by paired t-test. **D.** The boxplot describes the free triiodothyronine (FT3) and the triiodothyronine (T3) level of between DNP and DP samples at blood routine level. *P-value* was derived by permutation-based t-test. **E.** The correlation of FT3 and PA at blood routine level. **F.** The boxplots show the correlation of FT3 and lipid-related, glucose-related, and neuron-related biological processes, separately. The scatter size indicates the significance. *P-values* were derived from the permutation-based t-test. **G.** Bar plot depicting the spearman correlation of FT3 and different PA type of lipids including cholesterol, fatty acid, glycerophospholipid, ketone, and phospholipid.

As for the proteome characteristics alteration between DNP and DP

Having determined the altered clinical indicators between DNP and DP group, we next investigated the differences of plasma proteome profile between two groups. By comparing the ssGSEA scores between two groups (permutation-based t-test FDR <

0.05), we noticed the lipid-related processes was most up-regulated in the DNP group, followed by the neuron-related processes (**Figure RL 4A**). The upregulated lipid-related biological processes in DNP group included triglyceride rich lipoprotein particle remodeling, reverse cholesterol transport, lipoprotein lipase activity, lipoprotein particle clearance, and plasma lipoprotein assembly (**Figure RL 4B**). Considering the effect of BMI on the lipid-metabolism, we tested the BMI between the two groups and observed there's no significant difference (permutation-based t-test p -value = 0.60) (**Figure RL 4C**). Apolipoprotein family members, which can bind and transport blood cholesterol to various tissues of the body for metabolism and utilization, are of central importance in determining the response of immunotherapy. In our cohort, the plasma proteome provided us a comprehensive profile for different apolipoproteins. In detail, the level of high-density lipoprotein constituents APOC3, APOC2, and APOL1 increased in DNP group than DP group (permutation-based t-test p -value = 2.00E-4, 1.00E-3, and 7.70E-3) (**Figure RL 4D**). These results were consistent with that under the iBAQ quantification strategy.

The above analysis revealed the effects of DBT on the levels of APOC3, APOC2, and APOL1, which implied the potential roles of these proteins for monitoring the DBT response. In order to screen out the valuable proteins, we hypothesized that the ideal biomarker level should be consistently increased or decreased during the DBT for each patient. Therefore, we performed the time-series linear regression for 6 patients (P1, P16, P18 with no DPs and P2, P3, P22 with finally DPs) with more than 3 DBT cycles and set the DBT cycles as the covariable. Based on the hypothesis, we proposed a criterion to explore biomarkers, as for the patient with no DPs samples, (1) the expression level of the biomarker was gradually increased along with the therapy cycle and (2) the p -value < 0.05 in the linear model. As a result, APOC3 and APOC2 met the criterion (**Figure RL 4E**). Furthermore, between DNP and DP group, the expression level of APOC3 showed more significant difference than APOC2. Therefore, the

APOC3 was selected as the potential biomarker for indicating the DBT response state. Under the MaxLFQ quantification, the apolipoprotein-related biological process level and the APOC3 expression level were significantly higher in DNP as the results that revealed in the previous manuscript, which demonstrated its robustness for indicating the DBT response state. We have updated these related results from line 262 to line 271 and line 275 to line 277 in the revised manuscript.

Figure RL 4 A. The dot plot shows the difference of the gene set ssGSEA scores between DNP and DP samples. The dot size indicates the FDR of the adjusted permutation-based t-test p -values. The top related biological processes including lipid-related and neuron-related were annotated as red and orange. **B.** Bar plot depicting the significance of lipid-related biological processes in DNP samples. **C.** The box plot depicting the body mass index (BMI) between DNP and DP samples. P-values were derived by permutation-based t-test. **D.** The violin plots describe the level of apolipoproteins including APOC3, APOC2, and APOL1 between DNP and DP samples.

P-values were derived by permutation-based t-test. **E.** The panel shows the time series-linear regression of APOC3, APOC2, and APOL1 in 6 patients (P1, P16, P18 with all DNP samples, and P2, P3, P22 with DP samples). Dot color represents the evaluated response state.

As for the machine learning predictive model for the classification of DBT

We have identified clinical (PA, LDH) and proteome (APOC3) features present in the longitudinal cohort that associated with the response of DBT. This motivated the use of a machine learning pipeline (**Figure RL 5A**) to integrate the three features into a predictive model of DBT response to predict the disease progression state of patient. In this revised manuscript, we added more details for the pipeline and the final pipeline was consisted of seven parts including feature selection, feature combination, model benchmarking, model selection, parameter tuning, model refitting, and model evaluation (**2022, *Nature Medicine*, PMID: 35654907; 2022, *Nature*, PMID: 34875674**). Before the model construction, the collinearity of features was evaluated by the spearman analysis (**Figure RL 5B**).

A series of five therapy response prediction models including different feature combinations were derived including (1) PA, (2) LDH, (3) APOC3, (4) integrated clinical features (PA and LDH), (5) integrated all features (PA, LDH, and APOC3).

The models were based on multi-step processing pipeline (**Figure RL 5A**). Inside the pipeline, for each feature combination, we first compared different data preprocessing strategy (including the StandardScaler, MinMaxScaler, RobustScaler, and Normalizer) along with 21 state-of-the-art machine learning models and selected the top N (such as N = 5) best models. Next, a threefold cross-validation scheme was used to optimize model hyperparameters and the best model was selected as the final model. After determining the best model for each of the feature combinations, in order to evaluate

the model performance, we randomly split the cohort for 100 times to re-trained the model and re-evaluated the performance with more evaluation type including the area under the ROC curve (AUROC), balanced accuracy, F1, precision, recall, macro, and weighted evaluation metrics. We found the two models with the APOC3 as feature and the integrated all features showed better performance than that in other models with the clinical indicators (**Figure RL 5C**). Furthermore, in the model with integrated all features, LDH had the highest importance, followed by the PA and APOC3 during the model decision (**Figure RL 5D**).

Encouraged by these results, we wondered whether the usage of biological biomarker could spread from DBT to the ICIs monotherapy, especially in anti-PD1 monotherapy. We collected the independent validation cohort of anti-PD1 monotherapy consisted of 54 longitudinal samples from 27 patients and implemented the same plasma proteome profiling pipeline as the DBT cohort (one sample not pass the quality control during the MaxLFQ quantification and excluded for the further validation). For the five trained models, we evaluated the metrics on the independent validation cohort. For example, the AUROC, balanced accuracy, and F1 metrics were 0.576, 0.672, and 0.421 for the model with APOC3 as feature; 0.681, 0.592, and 0.308 for the LDH; 0.983, 0.853, and 0.750 for the PA; 0.932, 0.882, and 0.700 for the integrated clinical features; and 0.961, 0.944, and 0.762 for the integrated all features. As shown in **Figure RL 5E and 5F**, the model with integrated all features showed the better performance, followed by the model with the integrated clinical features. These indicated that PA, LDH, and APOC3 also have the potential to be predictive biomarkers for the response of anti-PD1 monotherapy. These results were consistent with that under the iBAQ quantification strategy.

In summary, after utilizing the MaxLFQ quantification strategy and the strict statistical method (the 70% valid values and the permutation-based t-test FDR), the main

conclusions remain unchanged such as the DBT has impact on the lipid, extracellular, and immune-related processes and the DBT has affected the liver function; the APOC3 expression level and the cholesterol-related biological processes level were significantly increased in DNP rather than in DP; the predictive model performance on the independent validation cohort (AUROC > 0.9). Notably, some non-main conclusion results were unrobust and excluded due to the MaxLFQ quantification strategy. For example, the FT3 most correlated with the glycerophospholipid followed cholesterol in the previous version, while most correlated with cholesterol in this version. This result indicated the MaxLFQ strategy excludes some noises and captures the key biological alterations across this cohort. We have updated these related results from line 328 to line 362 in the revised manuscript.

We thanked again for the reviewer's professional and construction suggestion. In this revision, we adopted the MaxLFQ strategy and updated the related information in **Figure, Result, Method, and Discussion** part. Furthermore, the related python codes for the MaxLFQ quantification and downstream analysis were uploaded on the GitHub (<https://github.com/Jiacheng-Lyu/DBT-plasma-proteome>).

Figure RL 5 A. Schematic of the machine learning pipeline for the predictive model of the patient response to the DBT. The pipeline includes seven parts as feature selection, feature combination, model benchmarking, model selection, parameter tuning, model refitting, and model evaluation. **B.** The heatmap depicting the collinearity of three features LDH, PA, and APOC3 used in the machine learning model on the discovery cohort. **C.** The different evaluation metrics including AUROC, balanced accuracy, precision, recall, F1, macro and weighted scores for models with different feature combinations. The metrics were evaluated by the different models that were refitted on the 100 different cohort splits. The color band indicates the 95% confidence interval. **D.** The feature importance for the model with the integrated all features. The error bar indicates the 95% confidence interval. **E.** The confusion matrix of the models with different feature combination on the independent validation cohort. **F.** The bar plot depicting the model performance on the independent validation cohort for the models with dif. The different evaluation metrics were used for the evaluation including AUROC, balanced accuracy, precision, recall, F1, macro and weighted scores.

Q3. Moreover, while the authors mentioned employing a total of 327 libraries as reference spectra libraries, the method section lacks sufficient details on the generation and utilization of this library. Does the 327 libraries derived from plasma proteome?

Response:

Thanks for the constructive comment. We apologize for the lack of description for the generation and utilization of this hybrid 327 library. In order to response the comments, we discussed the hybrid library construction pipeline from the three aspects as (1) the rationale of the hybrid library; (2) the design of the hybrid library; and (3) the advantage of the hybrid library for the DIA proteome data searching.

As for the rationale of the hybrid library

In order to construct the DBT plasma proteome profiling and explore the predictive biomarker candidates, we collected the longitudinal plasma samples from patients with advanced tumor who accept the DBT. The lower abundance proteins have the potential clinical significance, especially in the study of tumors (2016, *Cell Systems*, PMID: 27135364; 2023, *Nature Communications*, PMID: 37463882). It is indispensable to improve the plasma proteome identification to explore the diagnostic biomarker. Owing to the plasma protein were secreted from the tissue, the tissue proteome was the ideal library for the DIA plasma proteome. Meanwhile, due to the samples of this DBT cohort were from a phase I/Ib clinical trial, the different tumor types were enrolled in this cohort including lung cancer, cholangiocarcinoma, renal cell carcinoma, ovarian cancer, colorectal cancer, cervical squamous cell carcinoma, bladder cancer, and uterine corpus endometrial carcinoma. Hence, to improve the plasma proteome identification, designing corresponding libraries for different tumor types, that is the hybrid library, is a better strategy. Actually, in order to improve the proteome identification, the hybrid library-based DIA strategy has been developed and used in the DIA data analysis in the previous studies. For example, T.S. Zhu et al. generated a comprehensive pan-human library (2020, *Genomics Proteomics Bioinformatics*, PMID: 32795611) and G.

Rosenberger presented a generic large-scale human assay library (more than 10,000 proteins) to support protein quantification by SWATH-MS (a specific strategy of DIA). (2014, *Scientific Data*, PMID: 25977788). This library has been used for the DIA data quantification (2017, *Nature Methods*, PMID: 28825704; 2017, *Cell*, PMID: 28575672). Herein, to improve the proteome identification of the plasma samples for this pan-cancer cohort, we adopted the same strategy.

As for the design of the hybrid library

Up to now, we have published number of studies consisted more than 15 tumor types and up to 7,000 samples which focused on the multi-omics landscape of the different type of tumors, including colorectal cancer (CRCA) and stomach adenocarcinoma (STAD) (2023, *Gastroenterology*, PMID: 37995868; 2023, *Nature Communications*, PMID: 36788224), clear cell renal cell carcinoma (ccRCC) (2022, *Nature Communications*, PMID: 35440542), bladder cancer (BLCA) (2022, *Journal of Hematology & Oncology*, PMID: 35659036), esophageal squamous cell carcinoma (ESCC) (2023, *Nature Communications*, PMID: 36966136), cholangiocarcinoma (CHOL) (2023, *Hepatology*, PMID: 35716043), lung cancer (LC), and cervical squamous cell carcinoma (CESC) (2022, *Nature Communications*, PMID: 35851595) et al. Based on the accumulation of the proteome data of multiple tumor types, we selected the most heterogeneity samples with the highest protein identifications as the component of the hybrid library to maximize the coverage depth of the background library (Figure RL 6A).

Next, we performed the recommended library construction pipeline (2020, *Nature Methods*, PMID: 31768060; 2015, *Nature protocols*, PMID: 25675208; 2023, *Patterns*, PMID: 37521047). We processed all 327 of the newly acquired and collated proteomic datasets to generate the protein spectral libraries. The raw MS files underwent conversion to the mzML file format utilizing MSConvert software. For the

construction of consolidated spectral libraries, we engaged the FragPipe computational platform (version 12.1), equipped with MSFragger (version 2.2) (2017, *Nature Methods*, PMID: 28394336), Philosopher (version 2.0.0) (2020, *Nature Methods*, PMID: 32669682), and Python (version 3.6.7). The converted DDA mzML files, throughout the spectral library construction, were cohesively processed.

Peptide identification was executed from MS/MS spectra using FragPipe integrated with the MSFragger search engine. This was matched against the UniProt human protein database (reviewed sequences only; as updated on 2019.12.17, housing 20,406 entries), which also encompassed reversed protein sequences appended as decoys for ensuing false discovery rate (FDR) calculations. Technical specifications included both precursor and initial fragment mass tolerances set at 20 ppm, enabling spectrum deisotoping, mass calibration, and parameter optimization. We established enzyme specificity to ‘trypsin’ and permitted up to 2 missed trypsin cleavages. Configurations for peptide length ranged between 7 and 25, while peptide mass was set from 500 to 5,000 Da. Only precursor ion score charges of +2, +3, and +4 were considered. Cysteine carbamidomethylation was fixed as a modification, while variable modifications entailed N-acetylation and methionine oxidation. A maximum of three variable modifications per peptide was allowed.

Subsequent to the search, MSFragger search results (in pepXML format) were processed using the Philosopher toolkit. PeptideProphet (run with the high-mass accuracy binning and semi-parametric mixture modeling options) was employed to compute the posterior probability of correct identification for each peptide-to-spectrum match (PSM). The emergent output files from PeptideProphet were processed together using ProteinProphet to perform protein inference (assemble peptides into proteins) and synthesize a unified protXML file containing high-confidence protein groups. The combined ProteinProphet file was further processed using the Philosopher Filter

command, which characterized each identified peptide as a unique peptide to a particular protein (or protein group containing indistinguishable proteins) or assigned it as a razor peptide to a single protein (protein group) that had the most peptide evidence. Both unique and razor peptides were used for subsequent analysis. The data were filtered to 1% protein-level FDR using the picked FDR strategy. The peptide-, PSM-, and ion-level reports were then generated and filtered using the 2D FDR approach (i.e., 1% protein FDR plus 1% PSM/ion/peptide-level FDR for each corresponding PSM.tsv, ion.tsv, and peptide.tsv file). The resulting hybrid spectral library contained 215,529 precursors, representing 15,612 proteins. We have updated this related information from line 517 to line 560 in the revised manuscript.

Up to now, we have published some plasma proteomic studies which used this hybrid library, such as the upper tract urothelial carcinoma and colorectal tumor, et al (2023, *Cell Reports Medicine*, PMID: 37633276; 2024, *Nature Communications*, PMID: 38302471).

Figure RL 6A. The diagram of the sample composition and the library construction pipeline of the hybrid library used for the DIA MS data searching.

As for the advantage of the hybrid library for the DIA proteome data searching

After the 327-hybrid library construction, we evaluated the performance of this library and found it had the advantages for the coverage of the tumor biomarkers.

In order to evaluate the advantages of this 327-hybrid library, we constructed a library that consisted of only plasma samples. Meanwhile, we screened the published studies and collected a series 26 well-defined cancer biomarker. Particularly, DCP, HSP70, GGT1, and A1CF are predominantly served as biomarkers for hepatocellular carcinoma. CA9 and CD70 are regarded as specific biomarkers for renal cancer. ERBB2 (HER2) and AGR3 have been identified as distinct protein markers for gastric cancer. SOX2 and AGR2 have the potential to recognized the esophageal carcinoma. MUC2, EPCAM, CDX2, and MUC13 are considered as biomarkers for colorectal cancer. GFAP and NES are recognized biomarkers for glioma. In pancreatic adenocarcinoma (PAAD), MUC1, ANO1, and BASP1 are recognized biomarkers. Lung cancer is classified into non-small cell lung cancer (NSCLC) and small cell lung cancer (SCLC). Clinically, KRT5 and DSC3 have been identified as biomarkers for NSCLC, whereas CHGA and NCAM1 have been confirmed as markers for SCLC (**Table RL 1**).

We compared the coverage of the cancer biomarkers in the hybrid library and the only plasma library, separately. As shown in **Table RL1**, 25 out of 26 the cancer biomarkers were detected in the hybrid library, while only 12 out of 26 biomarkers were detected in the only plasma library. That is, the hybrid library could significantly expand the content, which especially benefit for the biomarker exploration.

Table RL 1

Tissue Type	Cancer Type	Reported Biomarker	Reference	PMID	Hybrid library	Only plasma library
Liver	Hepatocellular carcinoma	DCP	Journal of Hepatology, 2023	PMID: 37683735		

Liver	Hepatocellular carcinoma	HSP70	Journal of hepatology, 2009	PMID: 19231003	✓	✓
Liver	Hepatocellular carcinoma	GGT1	Gastroenterology, 2017; BMC Cancer, 2019	PMID: 28711626; PMID: 31455253	✓	✓
Liver	Hepatocellular carcinoma	A1CF	Cell reports, 2019; The JCI, 2021	PMID: 31597092; PMID: 33445170	✓	
Kidney	Kidney cancer	CA9	European urology, 2014; Cancer research, 1997	PMID: 24821582; PMID: 9230182	✓	
Kidney	Kidney cancer	CD70	Cancer research, 2006	PMID: 16489038	✓	
Esophageal	Esophageal carcinoma	SOX2	Human Pathology, 2012	PMID: 22748473	✓	
Esophageal	Esophageal carcinoma	AGR2	Human Pathology, 2012	PMID: 22748473	✓	
Gastric	Gastric cancer	ERBB2	Annals of oncology, 2008	PMID: 18441328	✓	
Gastric	Gastric cancer	AGR3	In vivo, 2023	PMID: 36593009	✓	
Colorectal	Colorectal cancer	MUC2	Gastroenterology, 2005;	PMID: 16285957	✓	
Colorectal	Colorectal cancer	EPCAM	British journal of cancer, 2014	PMID: 24786601	✓	✓
Colorectal	Colorectal cancer	CDX2	Annals of oncology, 2017	PMID: 28328000	✓	
Colorectal	Colorectal cancer	MUC13	Oncogene, 2019	PMID: 31427737	✓	
Pancreas	Pancreatic cancer	MUC1	Journal of gastroenterology, 2003	PMID: 14714254	✓	✓
Pancreas	Pancreatic cancer	ANO1	PNAS, 2019	PMID: 311825	✓	
Pancreas	Pancreatic cancer	BASP1	EBioMedicine, 2019	PMID: 30982764	✓	✓

Brain	Glioma	GFAP	Brain, 2007; Biosens Bioelectron, 2020	PMID: 17998256; PMID: 33160234	✓	✓
Brain	Glioma	NES	Cancer cell, 2020; Mol Neurobiol, 2017	PMID: 32396858; PMID: 26768429	✓	✓
Lung	Non-small cell lung cancer	KRT5	Cancer cell, 2022	PMID: 36368318	✓	✓
Lung	Non-small cell lung cancer	DSC3	Journal of thoracic oncology, 2011	PMID: 21623236	✓	
Lung	Non-small cell lung cancer	HSP90 AB1	Cell, 2020	PMID: 32649877	✓	✓
Lung	Small cell lung cancer	CHGA	Cell reports, 2020	PMID: 33086069	✓	✓
Lung	Small cell lung cancer	NCAM 1	Cancer cell, 2022	PMID: 36368318	✓	✓
Lymph	diffuse large B-cell lymphoma	CD10	Histopathology, 2001	PMID: 11493332	✓	✓
Lymph	diffuse large B-cell lymphoma	BCL6	Leukemia, 2007	PMID: 17625604	✓	

Q4. The overall conclusion drawn from the differential expression of APOC3 as a marker is based on a relatively small sample size of 22 and 27 patients, raising concerns about the robustness of the conclusions.

Response:

Thanks for the reviewer's professional comments. The DBT was an improved strategy for the monotherapy that can increase the immune response rates and the duration of response in patients with the advanced tumors. Up to now, there are a bunch of researches on the monotherapy, but lack on the DBT, particular in the research of the longitudinal cohort. Herein, in order to explore the predictive biomarkers that could

reflect the response state to the DBT, we collected the first longitudinal cohort in the worldwide consisting of a total of 113 proteome profiling of plasma samples from 22 patients.

According to the reviewer's suggestion, in order to ensure the robustness of the main findings, in this revision, we utilized the MaxLFQ quantification strategy along with the 70% valid value cutoff. Furthermore, we performed the strict statistical method. It is well-established that the permutation-based statistical method is a strategy to handle the small sample size problem. Hence, in this revised manuscript, we used the permutation-based t-test method to test the difference of clinical indicators and proteome data between DNP and DP samples. After applying the strict statistical method, the main results remain unchanged including the thyroid hormone related clinical indicators FT3 (*p-value* = 1.00E-4) and T3 (*p-value* = 5.40E-3) have the increased level in DNP rather than that in DP. Meanwhile, the apolipoprotein-related biological process level and the APOC3 expression level (*p-value* = 2.00E-4) could indicate the DBT response state (Details could be found in the response to **Q2**).

There are published clinical trials proved its clinical benefits for the patients with advanced tumor. Although the DBT is in ascendant for the immunotherapy, the predictive biomarkers for the response state to DBT is not developed, which limited its clinical practice. Herein, this study focused on this clinical problem and collected the longitudinal samples to explore the potential biomarkers, which has the clinical benefit for the patient who accept the DBT. Admittedly, due to this cohort was collected from the phase I/Ib clinical trial, this longitudinal cohort was hard to collection and the enrolled patient's number was limited. Although the rigorous statistical approach ensures the robustness of the main findings in a degree, we acknowledged that the relatively small sample size does have some potential impact on the robustness of the conclusions. Hence, these results can be served as a basis for the future work which

benefit for the clinical practice. In the future, we will continue to focus on the DBT plasma markers discovery and continue to collect larger cohorts to verify the findings. Meanwhile, in the revised manuscript, we added the limitation of the cohort size in the **Discussion** part (from line 441 to line 448). At the end, we thanked reviewer again for the attention to the patient number and robustness of the conclusions.

Q5. The description of the Maxquant and DIANN tools is incomplete, as the full reports of these searches were not provided. I obtained their submitted data from the specified link (<https://www.iprox.cn/page/PSV023.html?url=1698636112761nkY1>). The authors must present a comprehensive overview of their analysis tools for transparency and reproducibility.

Response:

Thanks for the comments about the description of the analysis tools including the MSFragger and DIA-NN. Actually, in this DIA-based proteome cohort research, we used the recommend pipeline to analyze the DIA data. The pipeline contained two steps as the DDA-based library construction and the DIA data analyzing.

As for the DDA-based library construction

The DDA data were searched against the UniProt human protein database (updated on 2019.12.17, 20412 entries) using FragPipe (v12.1) with MSFragger (2.2) (2017, *Nature Methods*, PMID: 28394336). Technical specifications included both precursor and initial fragment mass tolerances were set at 20 ppm. Up to two missed cleavages were allowed. The search engine set cysteine carbamidomethylation as a fixed modification and N-acetylation and oxidation of methionine as variable modifications. Precursor ion score charges were limited to +2, +3, and +4. The data were also searched against a decoy database to accept protein identifications at a false discovery rate (FDR) of 1%. A total of 327 libraries were used as reference spectra libraries. More details about the DDA library construction could be found in the response of the Q3 (part “*As for the*

design of the hybrid library”).

As for the DIA data analyzing

DIA data was analyzed using DIANN (v1.7.0) (2020, *Nature Methods*, PMID: 31768060) against the hybrid library. The DIANN search included the following settings: Precursor FDR: 1%, Log lev: 1, Mass accuracy: 20 ppm, MS1 accuracy: 10 ppm, Scan window: 30, Implicit protein group: genes, Quantification strategy: robust LC (high accuracy). Label-free protein quantifications were determined using an intensity-based, label-free approach incorporating delayed normalization and maximal peptide ratio extraction (MaxLFQ) approach. Peak area values were computed as components of their respective proteins.

This pipeline was deposited on the Firmiana which towards to a one-stop proteomic cloud platform for data processing and analysis (2017, *Nature Biotechnology*, PMID: 28486446). This quantification strategy was used for the published study (2023, *Cell Reports Medicine*, PMID: 37633276; 2023, *EMBO Molecular Medicine*, PMID: 37840432). Meanwhile, in this revision, to improve the reproducibility of the results and conclusions in this cohort study, the code of the downstream data analysis was deposited at GitHub (<https://github.com/Jiacheng-Lyu/DBT-plasma-proteome>). Thanks again for the reviewer point out the description of the analysis tools for transparency and reproducibility. We have updated the related information in the **Methods** part of the revised manuscript (from line 517 to line 560).

Q6. The authors' response in the method section, particularly in quantifying identified peptides, appears insufficient. The use of MaxLFQ values, which demonstrate better Coefficient of Variation (CoV), is not adequately justified compared to FOT or iBAQ. The clarification provided regarding the quantification process using iBAQ, FOT, and the associated normalization procedures lacks clarity and needs further elucidation.

Response:

Thanks for the reviewer's comments on the description of quantifying identified peptides and the comparison between iBAQ or FOT and MaxLFQ.

In this revised manuscript, according to the reviewer's professional suggestion, we utilized the MaxLFQ strategy to quantify the DBT and independent validation cohort proteome data, and reanalyzed the data. We found the main conclusions were remain unchanged including (1) the DBT has impact on the lipid, extracellular, and immune-related processes and the DBT affect the liver function; (2) the clinical indicator PA, increased in DNP, was most correlated with the cholesterol biological processes; (3) the APOC3 expression level and the cholesterol-related biological processes level were significantly increased in DNP rather than in DP; and (4) the predictive model performance on the test and independent validation cohort (AUROC > 0.9). The details could be found in the response to **Q2**.

In addition, we added the details about the quantification and associated data preprocessing (such as the normalization or the valid value cutoff) in the **Methods** part as

“

DIA data analyzing

DIA data were analyzed using DIANN (v1.7.0). The DIA-NN search included the following settings: Precursor FDR: 1%, Log lev: 1, Mass accuracy: 20 ppm, MS1 accuracy: 10 ppm, Scan window: 30, Implicit protein group: genes, Quantification strategy: robust LC (high accuracy). Label-free protein quantifications were determined using an intensity-based, label-free approach incorporating delayed normalization and maximal peptide ratio extraction (MaxLFQ) approach. Peak area values were computed as components of their respective proteins. The 'iq' R library was used for the MaxLFQ quantification.

Data preprocessing

Considering the balance between the confidence of protein identification and sample heterogeneity, the analysis in this study focused on the proteins identified in > 70% of samples. The missing value was served as NaN for the downstream analysis.

”

Q7. Furthermore, the data preprocessing methods deviate from proteomic community standards, as the manuscript currently allows for 30% valid values in total for both cohorts. I recommend revising this to adhere to the community standard of 70% valid values for enhanced reliability and process the downstream analysis.

Response:

Thanks for the reviewer's comments about the valid values cutoff which affected the reliability for the downstream analysis result.

Considering the balance between the confidence of protein identification and tumor intrinsic characteristic of heterogeneity, in previous published study, the less than 30% cutoffs were used (2018, *Nature Communications*, PMID: 29520031). This cohort was designed for the candidate biomarker exploring. Due to the sample size is relatively small, we fully agree with your suggestion that the 30% valid values could have some potential negative effects on the data analysis. Hence, in this revised manuscript, we used the 70% valid values to do the further downstream analysis for both discovery and independent validation cohort, which ensure and enhance the reliability and trustworthiness of the main findings. After used the 70% valid values, the main conclusions remain unchanged including (1) the DBT has impact on the lipid, extracellular, and immune-related processes and the DBT has affected the liver function; (2) the clinical indicator PA, increased in DNP, was most correlated with the cholesterol

biological processes; (3) the APOC3 expression level and the cholesterol-related biological processes level were significantly increased in DNP rather than in DP; and (4) the predictive model performance on the test and independent validation cohort (AUROC > 0.9).

The proteome data matrix was uploaded to the **Supplementary Table 1**, the description of the valid value was updated in the **Methods** part (from line 584 to line 586) and the analysis results were updated in the **Figure, Results, and Conclusion** parts in this revised manuscript. Thanks again for the reviewer's professional suggestion for enhance the reliability of the proteome data analysis.

Q8. The imputation strategy applied must also be thoroughly explained, and specific values should be provided.

Response:

Thanks for the comments about the missing value imputation. We apologize for not clarifying the descriptions of the methods section in the previous version. As for the imputation, which the reviewer pointed it out (“Considering the balance between the confidence of protein identification and sample heterogeneity, we selected proteins by specific threshold and then imputed them”), actually, this is a typo of the description.

In fact, in this cohort study, we have not used the any of the imputation strategy for the analysis of the clinical indicators and the proteome data. As for the statistical inference analysis, we used four types of analysis including the permutation-based t-test, pair-wised t-test, correlation analysis, and the univariate linear regression. These four statistical inferences were performed at the array level rather than the matrix level. Meanwhile, the imputation method could enroll some noise that change the distribution of data, which attenuate the statistical power. The published studies performed the statistical inference on the data without any of the imputation methods (2021, *Cancer*

Cell, PMID: 33417831; 2021, *Cell*, PMID: 34534465). Herein, we also performed the statistical inference without any of imputation methods, that is, leave missing value as NaN. As for the machine learning model construction, we excluded samples with missing values in any of the three features (LDH, PA, and APOC3). The related analysis python codes were uploaded on the GitHub (<https://github.com/Jiacheng-Lyu/DBT-plasma-proteome>).

In this revised manuscript, we updated this confused description as “Considering the balance between the confidence of protein identification and sample heterogeneity, the analysis in this study focused on the proteins identified in > 70% of samples. The missing value was served as NaN for the downstream analysis.” in the **Method** parts (from line 584 to line 586).

Q9. Concerns persist regarding the DIANN tool, which lacks a default option for iBAQ. The method section fails to specify how iBAQ values were extracted from the DIA-NN tool, necessitating clarification for transparency and reproducibility.

Response:

Thanks for the professional comments about the iBAQ value calculation. Actually, we deposit the DIANN tools on the Firmiana which towards to a one-stop proteomic cloud platform for data processing and analysis (2017, *Nature Biotechnology*, PMID: 28486446). The iBAQ values correlated well with known absolute protein amounts over at least four orders of magnitude and had a higher precision and accuracy than alternative measures of absolute protein abundance (2011, *Nature*, PMID: 21593866). In the previous version of manuscript, as the reviewer mentioned, the DIANN lacks a default option for the iBAQ, hence, the iBAQ value was calculated as dividing the total precursor intensities derived from the DIANN reports by the number of theoretically observable peptides of the protein. This quantification strategy was used for the published study (2023, *Cell Reports Medicine*, PMID: 37633276; 2023, *EMBO*

Molecular Medicine, PMID: 37840432).

In this revised manuscript, we used the MaxLFQ strategy to quantify the proteome data by the 'iq' R library that embedded the MaxLFQ quantification algorithm (<https://github.com/tvpham/iq>) (2020, *Bioinformatics*, PMID: 31909781). The related description was updated in the **Methods** part of the revised manuscript (from line 561 to line 570).

Q10. FDR permutation based two sample t-test (DNP vs DP, FDR 0.05) finds only 14 proteins significantly regulated in discover cohort (ACAN, HCY, APOC2, APOC3, AZGP1, CD44, CNDP1, F7, GFAP, HSPD1, IGHV1-69, IGHV2-70D, ITIH2, PAICS, RBP4) and no significance in validation cohort. The discrepancy between the number of significantly regulated proteins in the discovery cohort and the lack of significance in the validation cohort raises doubts about the reproducibility of the findings. Figures 3G, 4C, Supplementary Figure 5A, and Supplementary Figure 9B are flagged as not significant and, therefore, not reproducible.

Response:

Thanks for the comments about the reproducibility of the DBT cohort and the independent validation cohort. In order to response this comments more clearly, we split into three parts as (1) the comparison of the significantly differently expressed proteins between discovery cohort (DBT) and independent validation cohort; (2) the reproducibility of the result in the previous manuscript; (3) the performance of the machine learning model on the independent validation cohort.

As for the comparison of the significantly differently expressed proteins between discovery cohort (DBT) and independent validation cohort

In this revised manuscript, according to the comments Q2, Q7, and Q8, we utilized the MaxLFQ quantification strategy along with the 70% valid value cutoff and without

imputation to quantify both of the DBT and independent validation cohort proteome data and then, performed the permutation-based two sample t-test (DNP vs. DP, FDR cutoff set as 0.05). As shown in **Figure RL 7A**, there were 24 proteins increased in the DNP group for the DBT cohort and 19 proteins for the independent validation cohort (**Table RL 2**). Furthermore, we performed the over-represented analysis to reveal the main biological processes for the significantly expressed proteins in the two cohort. As shown in **Figure RL 7B**, the enriched biological processes in two cohorts including the complementary-related and cholesterol-related processes (BH adjusted *p-values* < 0.05). These results indicated the consistence of the proteome alteration in the discovery and independent validation cohort.

Table RL 2 DNP vs. DP in discovery cohort and independent validation cohort

DBT cohort				Independent validation cohort			
Genes	ttest statistics	ttest p-values	ttest FDR	Genes	ttest statistics	ttest p-values	ttest FDR
ADGR	4.375	1.00E-04	4.74E-03	AFM	4.971	2.00E-04	1.36E-02
B3		6.90E-03	3.55E-02	APOA4	2.982	5.20E-03	4.88E-02
AFM	2.740	1.00E-03	1.48E-02	APOM	3.184	2.30E-03	3.56E-02
APOC2	3.558	1.00E-04	4.74E-03	BTD	3.478	1.50E-03	2.50E-02
APOC3	4.286	7.70E-03	3.55E-02	F2	3.098	3.90E-03	4.34E-02
APOL1	2.728	9.00E-04	1.42E-02	GPLD1	4.358	5.00E-04	1.36E-02
BCHE	3.436	5.10E-03	2.93E-02	IGFALS	4.670	2.00E-04	1.36E-02
C1R	2.974	7.80E-03	3.55E-02	IGFBP3	3.351	3.40E-03	4.10E-02
CA1	2.753	7.60E-03	3.55E-02	IGHV1-3	3.254	2.70E-03	3.57E-02
CEP250	2.759	1.10E-02	4.57E-02	ITIH1	3.525	4.20E-03	4.34E-02
CLU	2.641	1.80E-03	1.78E-02	ITIH2	4.312	7.00E-04	1.52E-02
CNDP1	3.147						

F2	2.830	6.00E-03	3.31E-02	PON1	4.051	1.20E-03	2.17E-02
FETUB	2.626	1.05E-02	4.52E-02	RBP4	4.721	1.00E-04	1.36E-02
HBB	2.688	7.80E-03	3.55E-02	SERPI NA4	4.927	4.00E-04	1.36E-02
HBD	2.774	6.80E-03	3.55E-02	SERPI NA5	3.615	2.80E-03	3.57E-02
HGFA C	3.779	6.00E-04	1.28E-02	SERPI NC1	3.790	4.00E-04	1.36E-02
ITIH1	3.339	1.70E-03	1.75E-02	SERPI NF2	3.501	2.50E-03	3.57E-02
ITIH2	2.910	5.10E-03	2.93E-02	TF	3.967	7.00E-04	1.52E-02
KLKB 1	2.805	7.60E-03	3.55E-02	TMEM 256	3.581	1.10E-03	2.17E-02
RBP4	3.328	2.50E-03	2.12E-02				
SELEN OP	2.704	8.40E-03	3.76E-02				
SERPI NA4	2.491	1.23E-02	4.86E-02				
SERPI NA5	2.645	1.08E-02	4.57E-02				
TF	3.867	1.00E-04	4.74E-03				

As for the reproducibility of the result in the previous manuscript

In this revised manuscript, we utilized the MaxLFQ strategy to quantify the proteome data and the strict statistical method for the downstream analysis. After changing the strategy of data quantification and preprocessing, all main conclusions remain unchanged as (1) the DBT has impact on the lipid, extracellular, and immune-related processes; (2) the DBT has affected the liver function; (3) the thyroid hormone level and its positive correlation with cholesterol biological processes; (4) the increased level of apolipoprotein-related biological process and the APOC3 expression in DNP than that in DP group; and (5) the predictive model performance on the independent validation cohort (AUROC > 0.9). Some results are not robust such as the tubulin-

related biological processes, which could be served as the noise rather than the biological insights (corresponded to **Supplementary Figure 5A** in the previous version of manuscript), hence, we removed results of tubulin in the revised manuscript. Although the CTSL (a thyroid hormone related proteins) was not identified in this revision, we found apart from the clinical indicator free triiodothyronine (FT3), the triiodothyronine (T3) showed the increased level in DNP, which suggested the thyroid hormone plays roles during the DBT (**Figure RL 7C**, corresponded to **Figure 3G** in the previous version of manuscript), therefore, we updated results of T3 in the revised manuscript. Notably, we found the APOC3, APOC2, and APOL1 were significantly expressed between DNP and DP group, while the APOC4 was not (**Figure RL 7D**, corresponded to **Figure 4C** and **Supplementary Figure 9B** in the previous version of manuscript). We updated these results in **Figure 4C** of the revised manuscript.

In summary, based on the MaxLFQ quantification strategy and the strict statistical method, the key results and conclusion remain unchanged and the non-biological related noise was excluded in the research.

As for the machine learning model performance validation on the independent validation cohort

In this study, the independent validation cohort is also used for validating the performance of the machine learning-based predictive model that trained on the discovery cohort. In order to evaluate the model performance with different feature combinations (the combinations of the PA, LDH, and APOC3), we designed the benchmarking pipeline to perform the model construction (**Figure RL 7E**). As shown in **Figure RL 7F**, we evaluated the model performance of the five models on the independent validation cohort (corresponded to five different feature combinations), the confusion matrix suggested the integrated all features showed the best performance, followed by the integrated clinical features. This result is consistent with the result in

the previous version of manuscript that the integrated all features showed the best performance for distinguish DPs from DNPs, which suggested the APOC3 has the potential as the biomarker candidate.

At the end, we thanked reviewer again for improving this cohort study, which could more benefit for the clinical practice. The related information was updated in the **Figure (3D, 4C, 5D, and S8C), Results, Methods, and Supplementary Table 3, 4, and 5** in the revised manuscript (from line 328 to line 362 and line 561 to line 570).

Figure RL 7 A. The overlapped significantly expressed proteins (DNP vs. DP) in DBT cohort and independent validation cohort. **B.** The enriched biological processes of the DNP group in DBT cohort and independent validation cohort. **C.** The boxplot describes the free triiodothyronine (FT3) and the triiodothyronine (T3) level of between DNP and DP samples at blood routine level. *P-value* was derived by permutation-based t-test. **D.** The violin plots describe the expression level of apolipoproteins including APOC3,

APOC2, APOL1 between DNP and DP samples. *P-values* were derived by permutation-based t-test. **E.** Schematic of the machine learning pipeline for the predictive model of the patient response to the DBT. The pipeline includes seven parts as feature selection, feature combination, model benchmarking, model selection, parameter tuning, model refitting, and model evaluation. **F.** The confusion matrix of the models with different feature combination on the independent validation cohort.

REVIEWERS' COMMENTS

Reviewer #2 (Remarks to the Author):

The authors have responded to the comments merely adequate.